# A platform trial of neoadjuvant and adjuvant antitumor vaccination alone or in combination with PD-1 antagonist and CD137 agonist antibodies in patients with resectable pancreatic adenocarcinoma

A neoadjuvant immunotherapy platform clinical trial allows for rapid evaluation of treatment-related changes in tumors and identifying targets to optimize treatment responses. We enrolled patients with resectable pancreatic adenocarcinoma into such a platform trial (NCT02451982) to receive pancreatic cancer GVAX vaccine with low-dose cyclophosphamide alone (Arm A; $n = 16$), with anti-PD-1 antibody nivolumab (Arm B; $n = 14$), and with both nivolumab and anti-CD137 agonist antibody urelumab (Arm C; $n = 10$), respectively. The primary endpoint for Arms A/B - treatment-related change in IL17A expression in vaccine-induced lymphoid aggregates - was previously published. Here, we report the primary endpoint for Arms B/C: treatment-related change in intratumoral CD8+ CD137+ cells and the secondary outcomes including safety, disease-free and overall survivals for all Arms. Treatment with GVAX+nivolumab+urelumab meets the primary endpoint by significantly increasing intratumoral CD8+ CD137+ cells ($p = 0.003$) compared to GVAX+Nivolumab. All treatments are well-tolerated. Median disease-free and overall survivals, respectively, are 13.90/14.98/33.51 and 23.59/27.01/35.55 months for Arms A/B/C. GVAX+nivolumab+urelumab demonstrates numerically-improved disease-free survival (HR = 0.55, $p = 0.242$; HR = 0.51, $p = 0.173$) and overall survival (HR = 0.59, $p = 0.377$; HR = 0.53, $p = 0.279$) compared to GVAX and GVAX+nivolumab, respectively, although not statistically significant due to small sample size. Therefore, neoadjuvant and adjuvant GVAX with PD-1 blockade and CD137 agonist antibody therapy is safe, increases intratumoral activated, cytotoxic T cells, and demonstrates a potentially promising efficacy signal in resectable pancreatic adenocarcinoma that warrants further study.

✉ e-mail: lzheng6@jhmi.edu

Pancreatic ductal adenocarcinoma (PDA) has the highest case-fatality rate of any solid tumor. Even for the 15–20% of patients who are eligible for curative resection at diagnosis, 5-yr overall survival (OS) remains discouraging low at 20% with >80% of cases recurring within two years of definitive surgery[1]. While immune checkpoint inhibitors (ICIs) have dramatically changed frontline therapy and survival outcomes in a number of solid tumors, it has proven largely ineffective in patients with PDA[2,3]. A combination of low tumor mutation burden, deficient T cell activation, and an exclusive/suppressive tumor microenvironment (TME) act as barriers to antitumor immune responses against PDA. Utilizing a vaccine that induces and activates host effector T cells and co-administering it with immune-modulating agents that enhance antitumor T cell activity is a potential strategy for overcoming PDA resistance to ICIs.

Our group previously developed an allogenic, human granulocyte macrophage-colony stimulating factor (GM-CSF)-secreting whole-cell pancreatic cancer vaccine (GVAX) to promote T-cell responses against a range of tumor-associated antigens[4-7]. Along with this innovation, we constructed a perioperative resectable PDA clinical trial schema to evaluate early biologic responses to experimental therapies by testing one cycle of the experimental therapy 2 weeks prior to a planned surgical resection. This provided the opportunity to obtain a larger tumor specimen for a more comprehensive immune analysis of treatment-related changes in the heterogeneous TME[7]. In our initial studies, we reported that neoadjuvant treatment with GVAX alone was safe, feasible, and did not adversely affect surgical complication rates or survival in resected PDA patients[5,8]. We also reported that a single treatment of GVAX can induce the formation of tertiary lymphoid aggregates (TLAs) within the TME that function as local sites of T cell priming against PDA antigens. A higher density of TLAs was associated with longer OS[7-9]. Furthermore, PD-L1 expression was induced on both tumor epithelial cells and myeloid cells within the treated tumor TME, suggesting that vaccine therapy may prime PDA to respond to ICIs[7,10].

In order to rapidly identify the most critical T cell and immune microenvironment signals that require modulation, we conducted a platform neoadjuvant study (NCT02451982) that enrolls small numbers of patients into sequential treatment arms, each arm building on what is learned about the TME from prior treatment arms. We initially randomized the enrolled patients to either GVAX with low-dose cyclophosphamide (Cy-GVAX) (Arm A) or Cy-GVAX in combination with nivolumab, an anti-PD-1 antagonist mAb (Arm B) based on our first neoadjuvant GVAX clinical trial, as described above[7]. Prospectively banked paired baseline and on-treatment tumor biospecimens collected from the patients in these two arms (Cy-GVAX ± Nivolumab) demonstrated that an increase in the CD137+ activated T-cell subset in TLAs correlated with cytotoxic effector T cell signatures and was associated with improved 2-yr OS[9]. Notably, we observed that the CD137+ activated T cell subset within the TLA was low density and did not appear to infiltrate into the vicinity of neoplastic cells. This observation lead to our hypothesis that this effector T cell subtype may be expanded and mobilized by the use of a CD137 agonist treatment to generate a stronger antitumor response[11]. This hypothesis was further validated preclinically when an anti-CD137 agonist mAb combined with GVAX and an anti-PD-1 mAb significantly enhanced survival and correlated with increased intratumoral effector memory and cytotoxic T cells in a mouse model of PDA[12]. Taken together, CD137 was identified as a potential target for PDA immunotherapy (IO). Thus, we constructed and added a third treatment arm to our perioperative platform clinical treatment (Arm C) which combined urelumab, an anti-CD137 agonist mAb with Cy-GVAX and nivolumab.

CD137 (4-1BB; TNFSR9) is a T cell co-stimulatory receptor that mediates the activation of antigen-primed T cells with augmented survival, proliferation, and effector functions[13-15]. Mouse models had shown that agonist antibodies directed at CD137 led to complete rejections of transplanted tumors[16]. Urelumab (BMS-663513) is among the first few T-cell agonists that have been developed for therapeutic purposes[17]. A previous integrated evaluation of the safety data of urelumab showed significant transaminitis associated with doses of ≥1 mg/kg, but demonstrated to be safe with 0.1 mg/kg every 3 weeks as monotherapy and in combination with other IO agents[18]. Therefore, we chose to use a flat pharmacodynamically active dose of 8 mg in this study.

Here, we report the outcomes of the initial three treatments arms (Arms A, B, and C) of this study for patients with resectable PDA. These results demonstrate the safety of treating neoadjuvant and adjuvant Cy-GVAX with or without nivolumab and urelumab and the efficacy of treating with GVAX in combination with nivolumab and urelumab in significantly increasing tumor-infiltrating activated effector T cells and in improving disease-free survival compared to GVAX ± Nivolumab.

## Results

### Patient enrollment

From 3 March 2016 to 14 January 2019, 39 patients were randomized to Arms A and B (Figs. 1 and 2, Fig Supplementary Figure 1). From 15 February 2019 to 9 September 2020, Arm C enrolled 12 patients consecutively (Figs. 1 and 2, Supplementary Figure 1). During this time, randomization was held to ensure that Arm C met its accrual goal while urelumab was still available prior to its discontinuation by BMS (Supplementary Figure 1). In October 2020 the remaining supply of urelumab expired, and any Arm C patients who remained on study were transitioned to Cy-GVAX and nivolumab (Arm B regimen) with the same schedule and dosing (without urelumab). This impacted the treatment of three Arm C patients: the 1st received urelumab with their initial four study therapy cycles, the 2nd with their initial 2 cycles, and the 3rd with only their 1st (neoadjuvant) cycle. Once enrollment in Arm C was complete, randomized enrollment to Arms A and B restarted with an additional three patients enrolled, between 25 February 2021 and 10 September 2021, before the decision to close these respective Arms due to the plans to add new treatment Arms to the platform trial. The date of data cutoff for the final analysis for Arms A, B, and C was 25 May 2022.

Upon final analysis, 46 participants (n = 17 [A], n = 18 [B], n = 11 [C]) were enrolled and received the first dose of study treatment and were thus included in the safety cohort (Fig. 2). Forty (n = 16 [Arm A], n = 14 [Arm B], n = 10 [Arm C]) underwent subsequent, definitive (R0/R1) resection with surgical pathology confirming PDA and were thus evaluable for efficacy endpoints based on our pre-specified criteria (Fig. 2, Supplementary Note). Though the target number of evaluable patients in Arm A and B was not met, this did not inflate the type I error of the comparisons. Of the 40 evaluable patients, 37 went on to receive their 2nd treatment dose at the postoperative timepoint and standard of care (SOC) adjuvant therapy (patient patients were removed from trial for prolonged surgical recovery [>10 weeks] due to complications not related to the study drugs and another patient was removed due to metastatic progression found during recovery from surgery) (Fig. 2). During SOC adjuvant phase, one patient withdrew from treatment due to moving out of state, nine patients developed progressive disease, 1 patient died from an unknown cause, and 1 patient came off for grade 3 colitis. Twenty-five patients (n = 9 [Arm A], n = 9 [Arm B], n = 7 [Arm C]) received study treatments following the completion of SOC adjuvant therapy (Fig. 2). At the time of data cutoff, 23 patients (n = 7 [A], n = 9 [B], n = 7 [C]) completed all six priming treatments, and 9 (n = 2 [A] n = 2 [B] n = 5 [C]) entered and completed the extended-treatment phase (Fig. 2).

### Demographics and clinicopathologic characteristics

Demographics and tumor characteristics were similarly balanced across Arms A, B, and C (Table 1). Study participants evaluable for efficacy endpoints had a median age of 68 years old, with a majority having R0 resections (92.5%), pT-stage 2 disease (70%), moderate

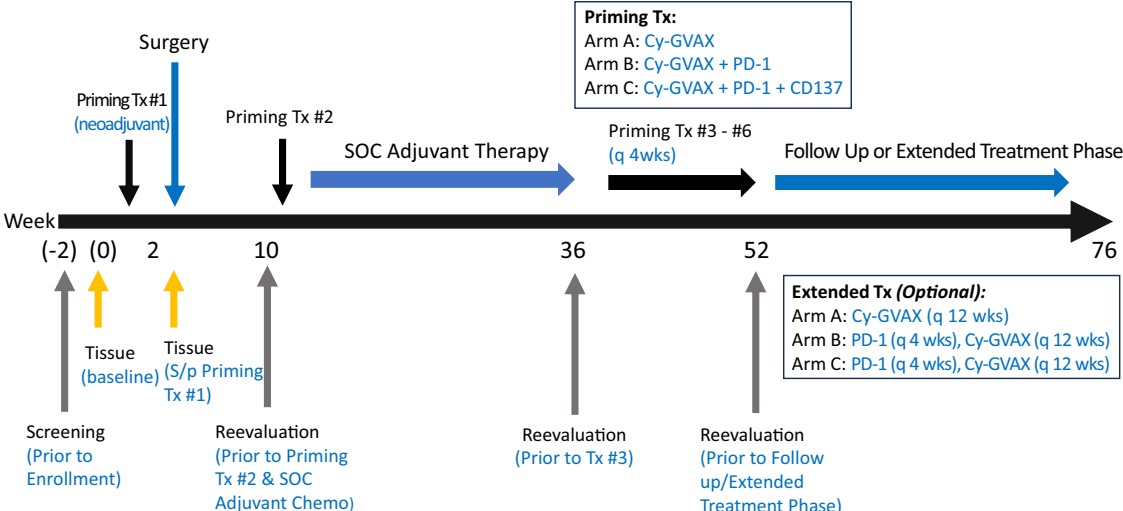

**Fig. 1 | J1568 study treatment schema.** Eligible patients with clinically resectable PDA received the first priming study treatment Cy-GVAX-based therapy (alone [Arm A], + PD-1 [Arm B], + PD-1 and CD137 [Arm C]) 2 weeks before the surgical resection, and the 2nd priming treatment 6–10 weeks following definitive surgical resection. Patients began SOC adjuvant therapy ~4 weeks following the 2nd study treatment. SOC adjuvant chemotherapy was administered as per the standard of care at the time at the discretion of the primary treatment oncologist. The 3rd (and up to 6th) priming study treatment was administered every 28 days beginning four weeks after the completion of SOC adjuvant therapy. Study treatment was given as follows: Day 1–Cyclophosphamide (Cy) 200 mg/m² IV (Arms A, B, C), nivolumab (PD-1) initially, 3 mg/kg, and later 480 mg IV following approval of every 4 week flat dose (Arms B, C), urelumab (CD137) 8 mg IV (Arm C Only); Day 2–GVAX intradermal (Arms A, B, C) was injected equally into six intradermal areas in both lower limbs and the non-dominant upper limb. This study began randomized enrollment to Arms A and B in March 2016. In October 2018, the study protocol was amended to add Arm C (due to limited supply of urelumab, Arm C had to enroll consecutively) as well as an optional "extended-treatment" phase. In this "extended-treatment" phase, all patients with no evidence of recurrence following the initial six priming doses of study treatment were given the option to receive additional Cy-GVAX every 12 weeks (up to 2 additional treatments), and, for Arm B and Arm C participants only, nivolumab (without ureulmab) every 4 weeks (up to six additional treatments).

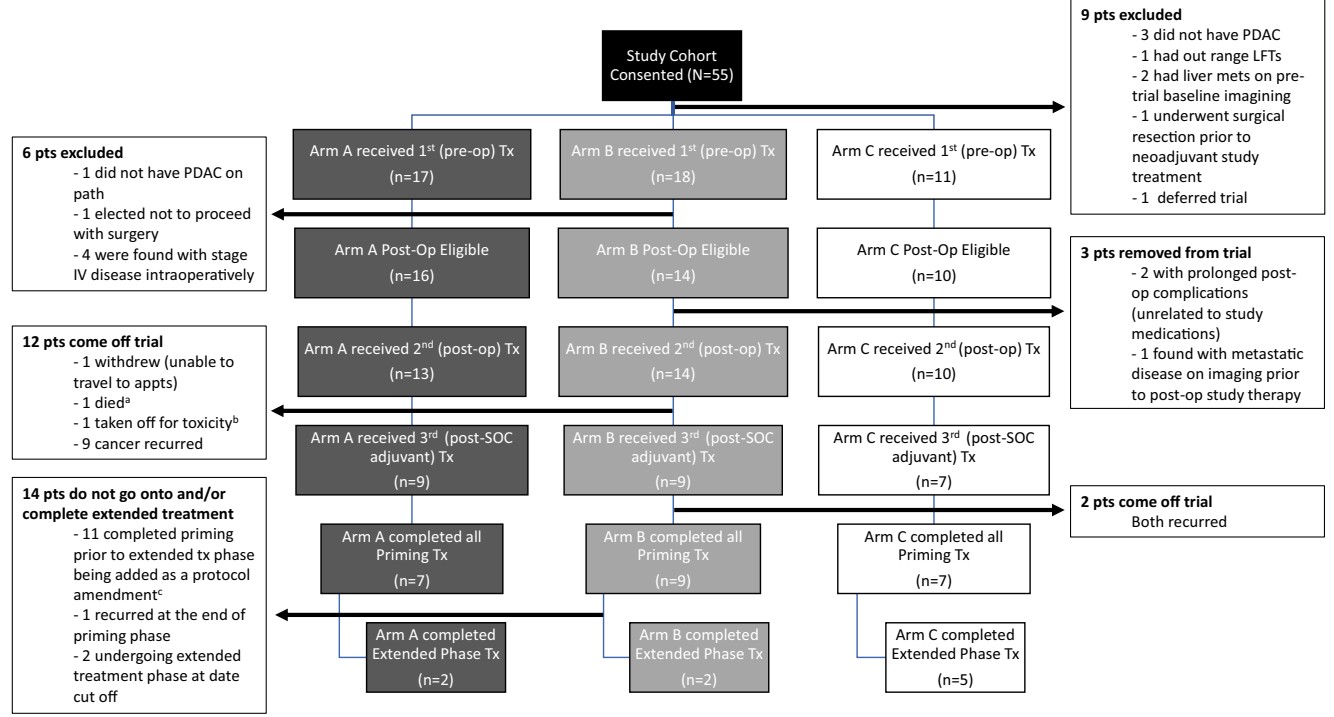

**Fig. 2 | CONSORT diagram of patient enrollment and on-study participation.** [a]Cause of death was unknown, occurred during standard of care adjuvant course, and was outside time range of reporting SAE; [b]Grade 3 immune-colitis; [c]protocol amendment with extended-treatment phase approved in October 2018.

(65%) or high (30%) tumor grade, and regional nodal involvement (70%) (Table 1). All patients enrolled had low tumor mutation burdens (median 0.94 mut/Mb [range 0–3.97]). While all arms had similar median adjuvant SOC therapy durations, patients in Arms A and B more often received gemcitabine (Gem) + capecitabine (Cap) as adjuvant chemotherapy (62.5% and 64.3%, respectively) while most of Arm C patients received adjuvant (m)FOLFIRINOX (70%) (Table 1). One patient in Arm A (6.3%) and 4 patients in Arm B received (28.6%)

## Table 1 | Baseline demographic and clinicopathologic disease characteristics (efficacy cohort [N = 40])

| Patient/disease characteristic | Arm A Cy-GVAX n = 16 | Arm B Cy-GVAX PD-1 n = 14 | Arm C Cy-GVAX PD-1 CD137 n = 10 |
|---|---|---|---|
| Age (yr) at surgery median (min, max) | 68.0 (47.0, 85.0) | 67.5 (53.0, 76.0) | 70.0 (46.0, 83.0) |
| Sex: female | 7 (43.8%) | 6 (42.9%) | 7 (70%) |
| Race | | | |
| white | 14 (87.5%) | 14 (100%) | 7 (70%) |
| Asian | 2 (12.5%) | 0 | 2 (20%) |
| Black | 0 | 0 | 1 (10%) |
| pT-Stage (AJCC 8th) | | | |
| 1 | 3 (18.8%) | 2 (14.3%) | 3 (30.0%) |
| 2 | 10 (62.5%) | 12 (85.7%) | 6 (60.0%) |
| 3 | 2 (12.5%) | 0 (0%) | 1 (10.0%) |
| 4 | 1 (6.3%) | 0 (0%) | 0 (0%) |
| pN-Stage (AJCC 8th) | | | |
| 0 | 5 (31.3%) | 4 (28.6%) | 3 (30.0%) |
| 1 | 2 (12.5%) | 5 (35.7%) | 4 (40.0%) |
| 2 | 9 (56.3%) | 5 (35.7%) | 3 (30.0%) |
| Tumor grade | | | |
| Well (1) | 1 (6.3%) | 1 (7.1%) | 0 (0%) |
| Moderate (2) | 9 (56.3%) | 10 (71.4%) | 7 (70.0%) |
| Poor (3) | 6 (37.5%) | 3 (21.4%) | 3 (30.0%) |
| Resection status: R0 | 16 (100%) | 12 (85.7%) | 9 (90.0%) |
| LVI: present | 11 (68.8%) | 8 (57.1%) | 6 (60.0%) |
| PNS: present | 14 (87.5%) | 13 (92.9%) | 9 (90.0%) |
| Adjuvant SOC chemo | | | |
| (m)FOLFIRINOX | 3 (18.8%) | 3 (21.4%) | 7 (70%) |
| Gem + Cap | 10 (62.5%) | 9 (64.3%) | 2 (20.0%) |
| Gem+ Nab-paclitaxel | 1 (6.3%) | 1 (7.1%) | 1 (10.0%) |
| Gem monotherapy | 1 (6.3%) | 2 (14.3%) | 1 (10.0%) |
| None | 3 (18.8%) | 0 (0%) | 0 (0%) |
| Time (days) from neoadjuvant study treatment to surgery | | | |
| Median (Q1, Q3) | 12.5 (11, 15) | 14 (12.3, 15) | 14 (11, 15.8) |
| Mo. of SOC Chemo median (Q1, Q3) | 5.26 (3.98, 6.34) | 5.29 (1.92, 5.86) | 5.34 (4.59, 5.74) |
| Adjuvant radiation | 1 (6.3%) | 4 (28.6%) | 0 (0%) |

*AJCC* American Joint Committee on Cancer, *LVI* lymphovascular invasion, *PNS* perineural spread, *SOC* standard of care, *(m)FOLFIRINOX* (modified) FOLFIRINOX (oxaliplatin + irinotecan + leucovorin, 400 mg/m² + infusional fluorouracil), *Gem* gemcitabine, *Cap* capecitabine.

received additional adjuvant chemoradiation (cRT), compared to 0 patients in Arm C (Table 1).

### Primary immunologic endpoints

The primary endpoint for Arms A and B were met by demonstrating tumor specimens from resected PDA patients treated with nivolumab and Cy-GVAX had significantly increased IL17A expression/TH17 infiltration in TLAs compared to tumor specimens from PDA patients treated with Cy-GVAX alone and was reported as part of correlative studies with Arms A and B, previously (19). For the biologic endpoint of CD8+ CD137+ tumor-infiltrating T cells, multiplex immunohistochemistry (mIHC) was performed on surgical specimens from 8 patients in Arm C. Tumors without an identifiable regions of interest (ROI) that contained epithelial neoplastic cells in the vicinity of TLAs were excluded from the analysis following the same standard established previously[9]. The results were analyzed and compared with those previously obtained from Arms A and B (n = 7 [A], n = 8 [B], n = 8 [C]).

As CD8+ CD137+ T cells were very rare on pre-treatment biopsy specimens[9], reporting the fold change between pre-treatment baselines and post-neoadjuvant immunotherapy samples would not be meaningful; therefore, this study only reported the density of CD8+ CD137+ T cells in the post-neoadjuvant immunotherapy PDA resected tumors.

Surgically resected tumor specimens from Arm C showed a significantly increased density of CD8+ CD137+ T cells within TLAs compared to those from Arm A (p = 0.007) and Arm B (p = 0.003), respectively (Fig. 3). Mean density of CD8+ CD137+ T cells within TLAs for Arms A, B, and C was 3.72%, 0.183%, and 27.9%, respectively. With a 152.5-fold difference in mean density of CD8+ CD137+ T cells within TLAs between Arms C and B, the primary endpoint was met. Using the overall median value (0.41% across arms) to stratify patients, >0.41% of CD8+ CD137+ T-cell density within TLAs correlated with improved disease-free survival (DFS) (HR 0.30, 95% confidence interval [CI] 0.11, 0.86, p = 0.026) (Supplementary Table 1, Supplementary Figure 2) but did not reach statistical significance with OS (HR 0.61 [95% CI 0.22,1.70], p = 0.349) (Supplementary Table 1, Supplementary Figure 2).

A Spearman correlation coefficient of 0.54 (Supplementary Figure 3) suggested a high correlation between CD8+ CD137+ T cells and CD8+Granzyme B (GZMB)+ T cells, a cytotoxic effector T cell subtype (p = 0.008). While CD8+ GZMB+ T cell density did not differ significantly across treatment arms (Fig. 3, Supplementary Table 1, Supplementary Figure 4) and did not correlate with survival (Supplementary Figure 4), we observed significantly increased CD8+ CD137+ GZMB+ T-cell density in TLAs in Arm C samples compared to specimens obtained from Arm A (p = 0.004) and B patients (p = 0.002) (Fig. 3). Using the overall median value (0.1% across arms) to stratify patients, >0.1% of CD8+ CD137+ GZMB+ T-cell density within TLAs, while not reaching statistical significance due to small sample size, was associated with a favorable HR for both DFS and OS compared to tumors with a CD8+ CD137+ GZMB+ T-cell density of 0.1% or less (DFS HR = 0.41[95% CI 0.14,1.17], p = 0.095; OS HR = 0.41 [95% CI 0.13,1.29], p = 0.127) (Supplementary Table 1, Supplementary Figure 5).

### Efficacy

At median follow-up times of 23.1 [Arm A], 26.1 [Arm B], and 31.6 [Arm C] months (mo), median DFS (95% CI) was 13.90 mo (5.59, NR), 14.98 mo (7.95, 44.09) and 33.51mo (16.76, NR) for Arms A, B, C, respectively (Table 2, Fig. 4). Detecting true statistical significance was limited due to the small number of patients within each treatment arm. In context of this, compared to Cy-GVAX alone (Arm A), adding nivolumab to Cy-GVAX (Arm B) did not improve DFS (HR 1.09 [95% CI 0.50, 2.40], p = 0.829) (Table 2, Fig. 4). Patients treated with the combination of urelumab, nivolumab, and Cy-GVAX (Arm C) demonstrated numerically-improved DFS when compared against those treated with Cy-GVAX alone (HR 0.55 [95% CI 0.21,1.49], p = 0.242) or Cy-GVAX with nivolumab (HR 0.51 [95% CI 0.19,1.35], p = 0.173) (Table 2, Fig. 4), but did not reach statistically significance. This favorable HR trend, though again not statistically significant, persisted after controlling for age, nodal spread, and adjuvant chemo regimen (HR = 0.64 [95% CI 0.19–2.19], p = 0.478 compared with Arm A; HR = 0.48 [95% CI 0.15–1.60], p = 0.232 compared with Arm B) (Supplementary Table 2).

Median OS (95% CI) was 23.59 mo (13.27, NR), 27.01 mo (20.76, NR), and 35.55 mo (17.74, NR) for Arms A, B, C, respectively (Table 2, Fig. 4). Compared to Cy-GVAX alone, adding PD-1 to Cy-GVAX did not improve OS (HR = 1.11 [95% CI 0.47, 2.63], p = 0.813) (Table 2, Fig. 4). Patients treated with the combination of CD137+ PD-1 + Cy-GVAX showed a numerically-improved OS when compared against those treated with Cy-GVAX alone (HR 0.59 [95% CI 0.18, 1.91], p = 0.377) and in combination with PD-1 (HR = 0.53 [95% CI 0.17, 1.67], p = 0.279) (Table 2, Fig. 4), but did not reach statistical significance. Similar to

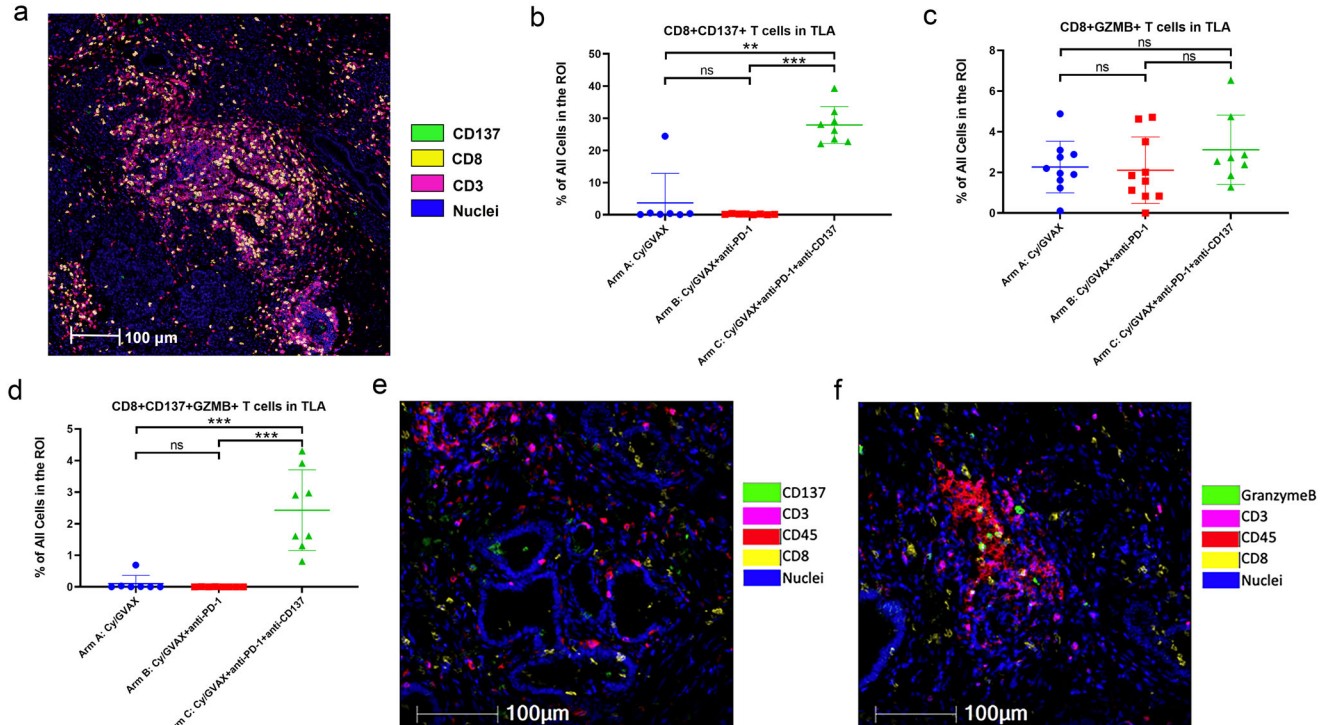

**Fig. 3 | Combination GVAX, Nivolumab, and Urelumab increase infiltrating CD3+ CD8+ CD137+ and CD3+ CD8+ CD137+ GZMB+ T Cells. a** Shown was one representative ROI that contains TLA and epithelial neoplastic cells in the vicinity; quantification was done within TLA and the tumor vicinity area outside TLA, respectively; mIHC marker pseudocolors: green = CD137, yellow = CD8, pink = CD3, blue = nuclei; **b** Comparison of the density of CD3+ CD8+ CD137+ T cells within the TLA among treatment arms as indicated. GVAX (Arm A) vs GVAX+PD-1+CD137 (Arm C): $p = 0.007$; GVAX+PD-1 (Arm B) vs GVAX+PD-1+CD137 (Arm C): $p = 0.003$. Arm A: $n = 7$; Arm B: $n = 8$; Arm C: $n = 8$. **c** Comparison of the density of CD3+ CD8+ GZMB+ T cells within TLA among treatment arms as indicated. Arm A: $n = 10$; Arm B: $n = 10$; Arm C: $n = 8$. **d** Comparison of the density of CD3+ CD8+ CD137+ GZMB+ T cells within TLA among treatment arms as indicated, GVAX (Arm A) vs GVAX+PD-

1+CD137 (Arm C): $p = 0.004$, GVAX+PD-1 (Arm B) vs GVAX+PD-1+CD137 (Arm C): $p = 0.002$. Arm A: $n = 7$; Arm B: $n = 8$; Arm C: $n = 8$. **e** Representative co-registered images of multiplex IHC showing CD3+ CD8+ CD137+ T cells within a tumor ROI; mIHC marker pseudocolors: green = CD137, pink = CD3, red = CD45, yellow = CD8, blue = nuclei. **f** Representative co-registered images of multiplex IHC showing CD3+ CD8+ GZMB+ T cells within a tumor ROI; mIHC marker pseudocolors: green = Granzyme B, pink = CD3, red = CD45, yellow = CD8, blue = nuclei. Two-sided Mann–Whitney were performed. Significance codes are displayed as follows: *<0.05; **<0.01; ***<0.001, ns = non-significance. All data shown as the mean ± SEM (standard error of the mean). Multiplex IHC analysis was repeated twice with consistent results.

DFS, this favorable HR persisted after controlling for age, nodal spread, and adjuvant chemo regimen (HR = 0.75 [95% CI 0.18, 3.10], $p = 0.692$ compared to Arm A; HR = 0.41 (95% CI 0.10, 1.62), $p = 0.202$ compared to Arm B) (Supplementary Table 3), but did not reach statistical significance.

On multivariate analysis (MVA), presence of nodal spread at time of surgery correlated with worse OS (HR = 2.92 [95% CI 1.02, 8.32], $p = 0.045$) and trended towards worse DFS (HR = 2.21 [95% CI 0.88, 5.53], $p = 0.091$) (Supplementary Table 2, Supplementary Table 3). Type of SOC adjuvant chemo regimen was not significantly correlated with DFS or OS in our study sample nor was tumor-stage (Supplementary Table 2, Supplementary Table 3, Supplementary Table 2, Supplementary Figure 6). Following one neoadjuvant dose of Cy-GVAX-based study therapy, 3 patients in Arm A (18.8%), 1 in Arm B (7.1%), 3 in Arm C (30%) displayed moderate pathologic responses (CAP grade 2)[19] upon surgical resection (Supplementary Figure 7). Due to confounding factors affecting a significant number of these patients including incomplete SOC adjuvant treatment courses, stage pT4 disease, and limited follow-up time, a correlation between pathologic response and survival could not be meaningfully assessed (Supplementary Figure 7).

To further address the potential confounder of SOC adjuvant chemo selection, Arm C patients were also compared to a matched-historical control cohort of resected PDA patients from the Johns Hopkins Pancreatic Cancer Registry during the time when Arm C was

enrolling. When matched 3:1 on adjuvant chemo regimen, age, and nodal disease status with propensity score matching (Supplementary Table 4, Supplementary Figure 8), Arm C patients reproduced a numerically favorable, although not statistically significant, HR for DFS compared to matched-historical controls: Arm C mDFS = 33.02 mo; Historical Control mDFS = 20.83 mo; stratified HR 0.72 [95% CI 0.29, 1.80], $p = 0.480$ (*DFS was measured starting the day of surgery for both groups, analysis stratified by adjuvant chemo type [FOLFIRINOX vs non-FOLFIRINOX]) (Supplementary Table 5, Supplementary Figure 8). We did not anticipate that this HR of DFS would reach statistical significance due to the small sample size in Arm C. Additionally, the DFS HR may have been underestimated due to the follow-up/surveillance imaging schedule being more stringent for patients on the trial compared to SOC DFS assessment in the historical cohort which carries a potential lead-time bias.

## Safety

All patients had mild (grade 1–2) vaccine injection site reactions such as local soreness, induration, erythema, and/or pruritus. One patient in Arm B had their treatment complicated by grade 3 immune-related diarrhea and colitis occurring while on treatment with SOC adjuvant FOLFIRINOX (Table 3). There were no other serious adverse events related to the study regimens (Table 3). In Arm C, 1 patient had a grade 3 rash that resulted in a one-time treatment delay and there was 1 instance of a grade 2 AST/ALT elevation that resolved without

**Table 2 | Disease-free survival and overall survival comparisons between treatment arms (efficacy cohort [N = 40])**

| Survival[a] | | Disease-free survival (months) | | | | Overall survival (months) | | | |
|---|---|---|---|---|---|---|---|---|---|
| Arm | Median follow-up (mo) | Events | Median (95% CI) | HR (95% CI) p value | | Events | Median (95% CI) | HR (95% CI) p value | |
| Cy-GVAX (n = 16) | 23.1 | 12 | 13.90 (5.59, NR) | Ref. | 0.92 (0.42–2.02) p = 0.829 | 12 | 23.59 (13.27, NR) | Ref. | 0.90 (0.38–2.14) p = 0.813 |
| Cy-GVAX PD-1 (n = 14) | 26.1 | 13 | 14.98 (7.95, 44.09) | 1.09 (0.50–2.40) p = 0.829 | Ref. | 11 | 27.01 (20.76, NR) | 1.11 (0.47–2.63) p = 0.813 | Ref. |
| Cy-GVAX PD-1 CD137 (n = 10) | 31.6 | 6 | 33.51 (16.76, NR) | 0.55 (0.21–1.49) p = 0.242 | 0.51 (0.19–1.35) p = 0.173 | 4 | 35.55 (17.74, NR) | 0.53 (0.17–1.67) p = 0.279 | 0.59 (0.18–1.91) p = 0.377 |

[a]Survival measured from date of first [neoadjuvant] study treatment; cox regression utilized. NR not reportable.

intervention (Table 3). There were no unusual patterns of post-operative complications, and there were no delays in surgery due to study regimen-related adverse events.

### Explorative immune analysis

Additional immune cell subtypes within TLAs following neoadjuvant immunotherapy in Arm C were analyzed (Supplementary Figure 9) and compared with previously reported results from Arms A and B[9]. The general CD8+ T cells increased in Arm B, but did not further increase in Arm C. Although PD-1+CD8+ T cells decreased in Arm B compared to Arm A as previously reported[9], PD-1+CD8+ T cells modestly increased in Arm C compared to Arm B likely as a result of T cell activation by CD137 agonist. Interestingly, Foxp3+CD4+T regulatory cells (Tregs) were significantly increased in Arm C compared to Arms A and B, consistent with the role of CD137 in Tregs as previously suggested[9]. Whether this induction of Tregs would suppress antitumor immune response remains to be investigated. Analysis of myeloid cell subtypes showed that the CD137 agonist decreased both M1 and M2-like tumor-associated macrophages, but did not change tumor-associated neutrophils significantly.

We also examined TIGIT+CD8+ T cells in TLAs in post-neoadjuvant immunotherapy tumors in Arm C (Fig. 5a) and found that higher density of CD137+ T cells in TLAs is associated in a trend with lower density of TIGIT + CD8+ T cells, supporting our previously developed hypothesis that CD137 agonist treatment may overcome T cell exhaustion[9]. Although the general CD8+ T cells in the TLAs did not increase in Arm C compared to Arm B, the percentage of CD137+ CD8+ T cells, but not GZMB+ CD8+ T cells, among CD8+ T cells significantly increased in Arm C. This seems to suggest that a subset of CD8+ T cells, likely a subset of GZMB+ cytotoxic T cells (considering their strong correlation with CD137+ CD8+ T cells [Supplementary Figure 3]) are converted to activated effector T cells following CD137 agonist treatment (Fig. 5b, c). As previously reported[9], CD8+ CD137+ T cells were essentially restricted in TLAs with minimal-to-no CD8+ CD137+ T cells seen in the vicinity outside the TLAs in Arms A and B. In contrast, this activated T-cell subtype made up 2–4% of cells in the tumor vicinity outside the TLAs within the same ROIs in PDAs from Arm C (Figs. 3e, 5d), suggesting that activated T cells may have migrated from TLAs to the vicinity of neoplastic cells.

## Discussion

This study demonstrates the feasibility of testing novel immunotherapy combinations in patients with resectable PDA using a platform clinical trial approach. In addition, this study design allowed for our team to uncover additional T-cell regulatory pathways activated in PDA through real-time correlative analysis of the first two experimental arms (Cy-GVAX ± nivolumab). Hypothesis-generating results raised by the correlative studies from the first two study arms subsequently informed the design of a third experimental arm (Arm C) where the CD137 agonist mAb, urelumab was added to Cy-GVAX+ nivolumab. The triplet combination met its primary endpoint: demonstrating promising tumor microenvironment changes by significantly increasing the percentages of tumor-infiltrating activated T cells (CD3+ CD8+ CD137+ T cells) and activated, cytotoxic effector T cells (CD3+ CD8+ GZMB+ CD137+ T cells). The observed treatment-related changes suggest that increasing the number of infiltrating effector T cells by itself may not be sufficient and that further optimization of effector T cell quality and activation, such as with an immune agonist mAb, may help enhance antitumor immune response to immunotherapy in PDA. Acknowledging that secondary clinical outcomes were limited by small sample size and imbalance in standard adjuvant chemotherapy regimens, the triplet regimen did demonstrate numerically-improved DFS in resected PDA patients. While this did not reach statistical significance, it merits further exploration for use in perioperative and post-adjuvant settings. These results support

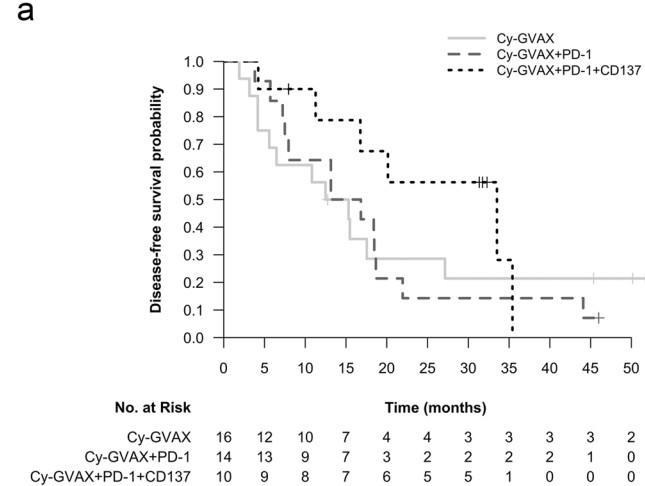

**Fig. 4 | Disease-free (DFS) and overall survival (OS) by treatment arm. a** DFS Kaplan–Meier curve stratified by treatment arm (efficacy cohort [*n* = 40]); **b** OS Kaplan–Meier curve stratified by Treatment Arm (Efficacy Cohort [n = 40]). Both DFS and OS were measured starting at time of first study therapy treatment. For

DFS, individuals were censored at the date of last restaging scan with documented disease status if they had no evidence of disease. For patients who died within 3 months of the last scan showing no recurrence, death was counted as an event. Otherwise, patients were censored at the time of last scan showing no recurrence.

the utility of our pilot trial with its platform design in which real-time correlative analysis in earlier study Arms can generate hypotheses that can then inform the rational selection of novel immunotherapies and therapeutic combinations to be tested in later Arms.

## Table 3 | Summary of study treatment-related adverse events[a] (safety cohort [*N* = 46])

| Treatment-related adverse events (TRAE) | Arm A Cy-GVAX *n* = 17 | Arm B Cy-GVAX PD-1 *n* = 18 | Arm C Cy-GVAX PD-1 CD137 *n* = 11 |
|---|---|---|---|
| TRAE, #pts (any grade) | | | |
| Abdominal pain | 0 (0%) | 0 (0%) | 3 (27.3%) |
| AST/ALT elevation | 0 (0%) | 0 (0%) | 1 (9.1%) |
| Chills/sweats | 4 (23.5%) | 2 (11.1%) | 2 (18.2%) |
| Dermatitis[b] | 3 (17.6%) | 3 (16.7%) | 6 (54.5%) |
| Diarrhea | 0 (0%) | 1 (5.6%) | 1 (9.1%) |
| Colitis | 0 (0%) | 1 (5.6%) | 0 (0%) |
| Dizziness/presyncope | 1 (5.9%) | 1 (5.6%) | 0 (0%) |
| Fatigue | 5 (29.4%) | 8 (44.4%) | 6 (54.5%) |
| Fever | 3 (17.6%) | 4 (22.2%) | 2 (18.2%) |
| Headache | 2 (11.8%) | 2 (11.1%) | 0 (0%) |
| Malaise | 1 (5.9%) | 5 (27.8%) | 1 (9.1%) |
| Myalgia/arthralgia | 1 (5.9%) | 6 (33.3%) | 2 (18.2%) |
| Nausea | 2 (11.8%) | 3 (16.7%) | 8 (72.7%) |
| Other | 0 (0%) | 0 (0%) | 2 (18.2%) |
| Swelling | 0 (0%) | 1 (5.6%) | 1 (9.1%) |
| Thyroid disorder | 0 (0%) | 0 (0%) | 4 (36.4%) |
| Vomiting | 2 (11.8%) | 2 (11.1%) | 2 (18.2%) |
| Grade ≥3 TRAE[c] | 0 (0%) | 1 (5.6%) | 1 (9.1%) |
| Serious TRAE (SAE)[d] | 0 (0%) | 1 (5.6%) | 0 (0%) |
| Trial therapy dose delay due to TRAE | 0 (0%) | 0 (0%) | 1 (9.1%) |
| Off trial due to TRAE | 0 (0%) | 0 (0%) | 0 (0%) |

[a]Does not include vaccine site reactions (VSR). Common VSRs included erythema, swelling, tenderness, and itching at vaccine sites.
[b]Includes hives, pruritus, rash.
[c]Grade 3+ TRAEs included grade 3 colitis (Arm B) and grade 3 rash (Arm C).
[d]SAE (treatment-related) was grade 3 colitis.

Patients randomized to Cy-GVAX alone or in combination with nivolumab, experienced mDFS and mOS intervals similar to those results established in phase III trials of their respective SOC adjuvant chemotherapy regimen (21–23) and were consistent with our previous trials of Cy-GVAX in the resectable PDA patient population (8, 24) (Supplementary Table 6). In the context of small sample size and imbalance in standard adjuvant therapy, we did observe a numerically, but non-statically significant, improvement in DFS for patients treated with the triple combination of Cy-GVAX, nivolumab, and urelumab. This, combined with the associated treatment-related increases of tumor-infiltrating activated effector T cells, may be a potential efficacy signal for this IO combination that should merit further study; particularly when placed in the context of DFS outcomes in previous adjuvant IO[5,7,20,21] and landmark phase III chemotherapy trials[22–24] in resectable PDA patients (Supplementary Table 6), While the mOS comparisons to these same were less favorable (e.g., PRODIGE mOS 54.4 mo)[22], it should be noted that conclusions about mOS may be of limited value given the small patient numbers, need for further follow up time to allow OS outcomes to mature, and, most significantly, influence of salvage therapy/subsequent lines of treatment rather than the study intervention. Because of this, DFS was favored as the more appropriate endpoint to evaluate the impact of the study intervention in this treatment setting.

While the observed immunologic and clinic outcomes are encouraging, specifically among the cohort that received the combination of Cy-GVAX, nivolumab, and urelumab, there are notable limitations to address and discuss. First, this trial was powered for a biologic endpoint rather than for clinical outcomes. Next, extrinsic factors required us to modify our intended study design. While we originally planned to expand treatment group randomization to 1:1:1 when Arm C was added 3 years into the trial, the finite supply of urelumab necessitated a pivot to enrolling Arm C patients consecutively to ensure accrual. As a result of this drug availability issue and a shifting paradigm of standard adjuvant treatment, patients in Arms A and B largely enrolled in 2016–2018 were more likely to have received adjuvant Gem-Cap, while Arms C patients enrolled during a period when (m)FOLFIRINOX was the preferred standard treatment. Additionally, standard nivolumab dosing changed while the trial was ongoing resulting in the majority Arm B patients receiving 3 mg/kg dosing rather than the 480 mg flat dose. Finally, this trial did not

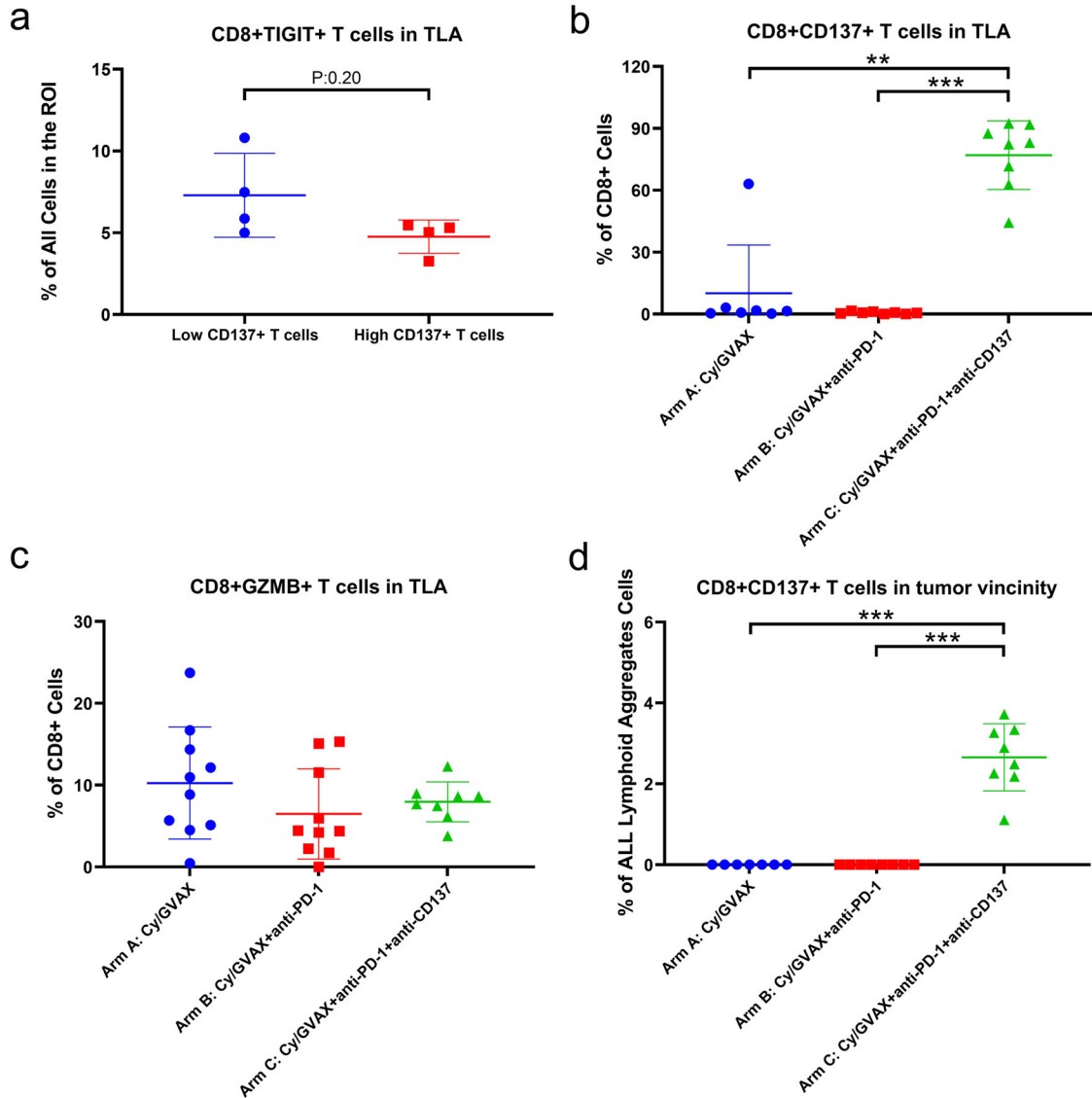

**Fig. 5 | Potential effects of CD137 agonist treatment on t cell exhaustion, activation, and trafficking. a** Samples in Arm C were subgrouped, according to the density of CD137+ CD8+ T cells in TLAs by using the mean of the density as cutoff, into two cohorts: low ($n=4$) vs. high ($n=4$) CD137+ T cells. The density of CD45+ CD3+ CD8+ TIGIT+ T cells was compared between the two cohorts. **b** The percentage of CD45+ CD3+ CD8+ CD137+ T cells among CD45+ CD3+ CD8+ T cells was compared between treatment arms. GVAX (Arm A) vs GVAX+PD-1+CD137 (Arm C): $p=0.0012$, GVAX+PD-1 (Arm B) vs. GVAX+PD-1+CD137 (Arm C): $p=0.0002$. Arm A: $n=7$; Arm B: $n=8$; Arm C: $n=8$. **c** The percentage of CD45+ CD3+ CD8+ GZMB+ T cells among CD45+ CD3+ CD8+ T cells was compared between treatment arms. Arm A: $n=10$; Arm B: $n=10$; Arm C: $n=8$. **d** The density of CD45+ CD3+ CD8+ CD137+ T cells in the tumor vicinity area outside TLAs, calculated as the percentage among all cells, was compared between treatment arms, GVAX (Arm A) vs GVAX +PD-1+CD137 (Arm C): $p=0.0003$, GVAX+PD-1 (Arm B) vs. GVAX+PD-1+CD137 (Arm C): $p=0.0002$. Arm A: $n=7$; Arm B: $n=8$; Arm C: $n=8$. All data shown as the mean ± SD. Treatment arms as indicated. Two-sided Mann–Whitney tests were performed; $p$ values were shown: *<0.05; **<0.01; ***<0.001; if not shown, non-significance. Multiplex IHC analysis was repeated twice with consistent results.

include patients who only received Cy-GVAX+ urelumab (without nivolumab) or Arms that received nivolumab and/or urelumab without Cy-GVAX. The justification for this was based on our group's in vivo work showing that the triplet combo of GVAX+PD-1+CD137 significantly improved survival compared to doublets of GVAX + CD137 and CD137+ PD-1[12]. While our group hypothesizes that combination GVAX, urelumab, and nivolumab together leads to robust and sustained antitumor immune responses via synergistic and/or complimentary mechanisms, future studies would benefit from a design that evaluates the contributive effects of GVAX (or an alternative antitumor vaccine platform) and ICI on immune agonist therapy.

It is important to acknowledge the higher percentage of Arm C patients receiving adjuvant (m)FOLFIRINOX compared to patients in

Arms A and B. To address this potential confounder, multiple strategies were employed to evaluate the additive contribution of IO triplet combination to the observed DFS trends. The multivariable survival analysis attempted to control for chemo regimen. Additionally, across all treatment arms, patients who received adjuvant (m)FOLFIRINOX appeared to have similar DFS when compared to the collective study participants who were treated with gemcitabine-based regimens. Finally, Arm C patients were also compared against a matched-historical control cohort. There are clear limitations of this study and the above analyses driven largely by the sample size. However, the early clinical and immune response signals observed in this small cohort support a follow-up, randomized, phase 2 study designed and powered to assess the clinical efficacy of the triple IO combination used in Arm C.

In summary, treatment with GVAX (with low-dose Cy) alone or in combination with PD-1 blockade and CD137 agonist mAb was feasible and safe in patients with resectable PDA treated in the neoadjuvant and adjuvant settings. The combined regimen of Cy-GVAX, Nivolumab, and Urelumab was well-tolerated, increased TME immunologic responses, and demonstrated a potentially promising efficacy signal meriting further validation in a larger, randomized clinical trial. Additional biomarker studies are warranted, particularly on the immunosuppressive TME and T cell exhaustion pathways, to inform new Arm design for our platform trial.

## Methods

### Ethics and compliance

The study was approved by the Johns Hopkins Institutional Review Board (IRB) and Institutional Biosafety Committee (IBC), as well as the FDA Center for Biologics Evaluation and Research and the National Institutes of Health Recombinant DNA Advisory Committee (J1568, NCT02451982). The trial was conducted according to the Declaration of Helsinki and the Good Clinical Practice guidelines of the International Conference on Harmonization. Informed, written consent was obtained from all patients.

### Study design

This is a multi-arm, open-label, pilot platform study of patients with PDA who were scheduled to undergo pancreaticoduodenectomy at the Johns Hopkins Hospital (Baltimore, Maryland, USA). Eligible patients with resectable PDA received GVAX administered intradermally in combination with immunomodulatory doses of cyclophosphamide (Cy), with or without PD-1 inhibitor mAb (nivolumab, BMS-936558) and CD137 agonist mAb (urelumab, BMS-663513) for neoadjuvant and adjuvant treatment in addition to standard (SOC) adjuvant chemotherapy (chemo) and/or chemoradiation (cRT) therapy at specified intervals. Further information regarding the platform trial (or trial design) can be found at clinicaltrials.gov.

### Study population and evaluable patients

The target population for this study was patients with resectable PDA. Key eligibility requirements at time of enrollment included the following: suspected or confirmed diagnosis of pancreatic ductal adenocarcinoma; deemed to be surgically resectable by multidisciplinary tumor board review; no known second malignancies within five years of diagnosis of pancreatic cancer; Eastern Cooperative Oncology Group (ECOG) performance status ≤1; no radiographic evidence of metastases; no serious autoimmune disease requiring treatment with systemic corticosteroids; adequate organ function. All patients who met trial criteria were offered the trial without any anticipated bias. Additional eligibility requirements for trial continuation following surgery included: confirmed histologic diagnosis of pancreatic ductal adenocarcinoma, R0 or R1 resection, surgical recovery by post-op week 10, and no evidence of distant metastases. As pre-planned in the clinical protocol, patients evaluable for efficacy endpoints were those who received at least one dose of the study drug, followed by definitive surgery with a pathological diagnosis of pancreatic ductal adenocarcinoma (or histologic subtype). Patients with R2 resections or with distant metastases were excluded. All subjects who received the first dose of study therapy were evaluable for safety endpoints.

### Outcomes/endpoints

The primary endpoint for all the arms was biological/immunologic endpoints. This platform trial initially has two arms, A and B. The primary endpoint for Arms A and B was IL17A expression in vaccine-induced lymphoid aggregates in resected PDAs from patients treated with the combination of Cy-GVAX with or without nivolumab[9]. The primary endpoint for Arms A and B was previously analyzed; and Arm

C was subsequently added as a result of additional correlative studies with Arms A and B[9]. Such correlative studies also determined the primary biologic endpoint for Arm C as the CD8+ CD137+ T-cell density within tumor regions of interest (containing at least one TLA) in surgically resected specimens following neoadjuvant immunotherapy.

This study was powered for the above primary biologic endpoints. The sample size of evaluable subjects for the respective treatment groups ($n = 17$ [Arm A], $n = 17$ [Arm B], $n = 10$ [Arm C]) provided an 82% power (based on two-sample $t$ test on log-transformed values) to detect a 2.2-fold difference in IL17A expression levels in TLA between resected tumor specimens from Arms A and B after neoadjuvant immunotherapy and an 89% power to detect threefold difference in intratumoral CD8+ CD137+ cells in Arm C resected PDAs compared to Arm B following neoadjuvant study treatment, with two-sided type I error of 0.05 (Supplementary Note). The effect size was projected based on correlative studies with Arm A and B[9]. Since both primary comparisons−1) comparing IL17A expression between Arms A and B, and 2) comparing CD137+ T-cell density between Arm C and B−were each of respective interest, they were not subjected to the multiple comparison adjustment. Secondary outcomes included the clinical endpoints of DFS and OS. Additional exploratory correlative studies were also conducted.

### Enrollment

From 3 March 2016 to 14 January 2019, eligible patients were enrolled and randomized 1:1 into the initial two treatment arms, Cy-GVAX (Arm A) or Cy-GVAX plus nivolumab (Arm B) (Supplementary Figure 1). Randomization was stratified by age of enrollment (≤65 and >65 years old). In October 2018, a 3rd treatment arm (Arm C) was added for patients to receive Cy-GVAX plus nivolumab and urelumab (Supplementary Figure 1). Discontinuation of urelumab production necessitated enrolling Arm C patients consecutively in order to meet the Arm C accrual goal (15 February 2019−9 September 2020). Following the completion of Arm C accrual, enrollment to Arms A and B resumed in a randomized fashion (25 February 2021−10 September 2021). Due to plans to add new treatment Arms for this patient population, Arms A and B were then closed (4 November 2021) and a final analysis was conducted. The data reported herein reflect follow-up through 25 May 2022.

### Treatment schema and assessments

Study treatment was given as follows: Day 1−Cyclophosphamide (Cy) 200 mg/m$^2$ IV (Arms A, B, C), nivolumab (PD-1) initially, 3 mg/kg, and later 480 mg IV following approval of every 4 week flat dose (Arms B, C), urelumab (CD137) 8 mg IV (Arm C Only); Day 2−GVAX intradermal (Arms A, B, C) was injected equally into six intradermal areas in both lower limbs and the non-dominant upper limb, as described previously[20].

After determination of clinical resectability (based on collective multidisciplinary expert opinion) and obtaining consent, patients received the first priming treatment Cy-GVAX-based therapy (alone [Arm A], + PD-1[Arm B], + PD-1 and CD137[Arm C]) 2 weeks before the surgical resection, and the 2nd priming treatment 6−10 weeks following definitive surgical resection. Patients began adjuvant chemotherapy ~4 weeks following the 2nd study treatment (Fig. 1). SOC adjuvant chemo was administered as per standard of care but modified as needed at the discretion of their primary oncologists. Treatment with SOC cRT was determined by each primary oncology team. The 3rd (and up to 6th) priming study treatment was administered every 28 days beginning 4 weeks after the completion of SOC adjuvant chemotherapy and/or radiation (Fig. 1). Beginning in August 2018, the study protocol was amended to include an "extended-treatment"

phase: following the initial 6 priming doses of study treatment, all patients with no evidence of recurrence were given the option to receive additional Cy-GVAX every 12 weeks (up to 2 additional treatments). During the "extended-treatment phase," Arm B and Arm C participants also received nivolumab (without ureulmab) every 4 weeks for up to 6 additional treatments if response/tolerance persisted (Fig. 1).

Patients were assessed for local or distant disease recurrence by CT scan prior to enrollment, following recovery from definitive surgery, and then every 2 months while on-study treatment. All CT scans were read by Johns Hopkins Radiology attendings and discussed further at our institutional multidisciplinary tumor board if further diagnostic clarification was needed. Toxicities were graded using CTCAE v.5. Postoperative complications within 30 days of surgery were graded by the Clavien Dindo Criteria[25].

### Multiplex Immunohistochemistry (mIHC)

Primary PDA tumor samples were obtained from endoscopic ultrasound-guided fine needle core biopsies (EUS-FNB) or surgically resected tumors. Both fresh tissue and formalin-fixed paraffin-embedded (FFPE) tissue blocks were obtained. Sequential staining-striping mIHC protocol was used[10]. 5-μm thick FFPE tissue sections slides were stained by hematoxylin and scanned using NanoZoomer (Hamamatsu). Following antigen retrieval by microwave treatment with the Antigen Retrieval Citra, sequential multiple iterative IHC cycles involving staining, scanning, and antibody/chromogen stripping, was performed[10]. Detailing of the primary antibodies, incubation times, horseradish peroxidase (HRP)-conjugated polymer, aminoethylcarbazole reaction time for chromogenic detection was described in Supplementary Table 7[10]. Negative control images were obtained after the last antibody and chromogen stripping. The first staining panel, which was designed for CD8+ T-cell markers, included CD45, CD3, CD8, PD-1, CD137, and GZMB. The second staining panel, which was designed for myeloid cell markers and Treg markers, included CD45, CD3, CD8, CD4, CSF-1R, CD68, CD163, CD66b, Foxp3, and TIGIT. Digitized images obtained with NanoZoomer were co-registered via the specific CellProfiler (version 2.1.1) pipeline[10]. With the assistance of pathologist (E.T.), tumor areas were identified on the H&E slides. Three rectangle ROIs of ~3000 × 3000 pixels each containing one TLA and epithelial neoplastic cells in the vicinity, known to be representative of the larger whole tumor based on prior study[10], were chosen for analysis. Immune cell subtypes were defined by multiple immune cell markers in consistency with prior studies[9]. Immune cell density was defined as the percentage of the specific immune cell subtype among all cells within TLAs in consistency with prior studies[9]. Multiplex IHC analysis was repeated twice with consistent results.

### Statistical considerations

Descriptive statistics were used to describe the study population and adverse events.

DFS was defined as time from the start of study treatment until radiographic recurrent disease and/or death. Individuals were censored with respect to DFS at the date of last restaging scan if they had no evidence of disease. For patients who died within 3 months of the last scan showing no recurrence, date of death would be counted as an event date of DFS. Otherwise, they would be censored at the time of last scan showing no recurrence. OS was defined as the time from study treatment start until death, regardless of cause. Comparisons of the OS and DFS between the treatment arms were made using Kaplan–Meier curves and log-rank tests. The Cox model was used to estimate hazard ratio. Multivariate Cox regression was implemented to compare study treatment arms adjusting for age at surgery, nodal disease status, and adjuvant SOC chemotherapy ([m]FOLFIRINOX vs. other). In addition to cross-treatment arm comparison, nearest neighbor propensity score matching, in combination with exact matching, were used to analyze study treatment efficacy between Arm C patients and an institutional historical cohort. Mann–Whitney tests were used for comparison of immunologic endpoints between treatment groups. The association between immunologic endpoints was quantified using the Spearman correlation coefficient. Tests were two-sided and $p$ values <0.05 were considered to indicate statistical significance. Study data were recorded using Microsoft Excel (2016) and Microsoft Access (2016) and stored on a HIPAA complaint drive. Statistical analyses were performed using R (version 4.2.1) and Prism (version 9.3.1). Figures were composed using R, Microsoft Powerpoint (2016), and Prism.

### Historical control cohort generation and analysis

The historical cohort consists of 48 patients who received definitive pancreatectomy surgery from Jan 2018 onwards at Johns Hopkins Hospital/Sidney Kimmel Cancer Center. The data cutoff used for the historical cohort analysis is 5/25/22. Eligibility criteria for the historical cohort were patients who underwent upfront definitive (R0/R1) pancreatoduodenectomy and excluded patients who had received neoadjuvant therapy, had T4 and/or M1 disease, and those who had undergone distal pancreatectomy. These historical cohort patients were matched with Arm C patients based upon age, nodal involvement at time of surgery, and adjuvant chemotherapy regimen (either [m]FOLFIRINOX or non-[m]FOLFIRINOX). Matching was done without replacement (controls are only allowed to be used as a match once). We implemented nearest neighbor propensity score matching with a matching ratio 3:1 (control to treated). The grouping variable is control (historical cohort) vs treated (Arm C) and the variables being matched on are age, nodal disease status, and adjuvant chemotherapy, where exact matching was specified for adjuvant chemotherapy. The resulting groups were well balanced with an absolute standardized mean difference below 0.2. Cox regression (measured from date of surgery for both historical control cohort and Arm C) was performed for DFS to compare Arm C vs. matched-historical, stratified by adjuvant chemotherapy.

### Reporting summary

Further information on research design is available in the Nature Portfolio Reporting Summary linked to this article.

## Data availability

Disaggregated data at the individual patient level (Supplementary Data 1) and source data for Figures and Tables (Source Data file) are provided. The study protocol is available as Supplementary Note in the Supplementary Information file. All data except the original multiplex IHC staining images are available within the Article, Supplementary Information, or Source Data file. The original multiplex IHC staining images are available from the corresponding author upon request. Additional individual de-identified participant data could be shared upon request from the corresponding author. Additional details of the trial, data, contact information, proposal forms, and review and approval process are available at the following website: https://clinicaltrials.gov/ct2/show/NCT02451982. Source data are provided with this paper.

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

## Acknowledgements

This study was supported by NIH grant R01 CA169702 (L.Z.); NIH grant R01 CA197296 (L.Z., E.J.); NIH grant P01CA247886 (E.J., L.Z., D.L.); NIH grant P50 CA062924 (E.J., L.Z.); NIH grant 5T32CA009071-38 (T.H.); NIH grant K12-CA090625-17 (T.H.); a Bristol–Myers Squibb (BMS) grant (E.J., L.Z.); the Viragh Foundation and the Skip Viragh Pancreatic Cancer Center at Johns Hopkins (E.J., L.Z.); the Pancreatic Cancer Precision Medicine Center of Excellence Program at Johns Hopkins (L.Z.); the Sidney Kimmel Comprehensive Cancer Center Grant P30 CA006973, and the Lustgarten Pancreatic Cancer Convergence Grant (E.J.). We thank BMS for their grant support and for supplying study medications. The sponsor did not have a direct role in study design, data collection, and analysis, or manuscript writing.

## Author contributions

L.Z. and E.J. contributed to the conception, design, and planning of the study. T.H., C.J., J.H., R.P., H.C., K.L., J.G., B.C., Q.Z., T.M., J.D., R.K., D.L., A.D.J.-A., D.T.L., T.Z., A.N., R.A., R.B., W.B., K.S., C.W., E.T., K.P., J.H., L.Z. acquired the data. T.H., S.L., H.W., K.L., J.G., B.C., Q.Z., R.A., E.T., J.H., L.Z. analyzed the data. C.J., S.L., H.W., J.H., R.P., H.C., J.D., R.A., A.N., D.L., A.D.J.-A., D.T.L., R.K., R.B., W.B., K.S., C.W., E.T., E.J., J.H. contributed to the interpretation of the results. T.H. drafted the manuscript. T.H., S.L., and K.L. collaborated on the manuscript tables and figures. All authors contributed to the drafting or critical review of the manuscript and all authors had access to the data, which was verified by L.Z., T.H., T.M., J.D., R.P., K.L.

## Competing interests

L.Z. receives grant support from Bristol-Meyer Squibb, Merck, Astrazeneca, iTeos, Amgen, NovaRock, Inxmed, and Halozyme. L.Z. is a paid consultant/Advisory Board Member at Biosion, Alphamab, NovaRock, Ambrx, Akrevia/Xilio, QED, Natera, Novagenesis, Snow Lake Capitals, BioArdis, Amberstone Biosciences, Tempus, Pfizer, Tavotek Lab, ClinicalTrial Options, LLC, and Mingruizhiyao. L.Z. holds shares at Alphamab, Amberstone, and Mingruizhiyao. E.J. reports other support from Abmeta and Adventris, personal fees from Achilles, Dragonfly, Mestag, The Medical Home Group, and Surgtx, other support from Parker Institute, grants and other support from the Lustgarten Foundation, Genentech, BMS, and Break Through Cancer outside the submitted work. D.T.L. serves on advisory boards for Merck, Bristol Myers Squibb, Nouscom, G1 Therapeutics, Janssen, and Merus and has received research funding from Merck, Bristol Myers Squibb, Curegenix, Nouscom, and Abbvie. She has received speaking honoraria from Merck and is an inventor of licensed intellectual property related to technology for mismatch repair deficiency for diagnosis and therapy (WO2016077553A1) from Johns Hopkins University. The terms of these arrangements are being managed by Johns Hopkins. R.A. receives grant support from Bristol-Meyer Squibb, RAPT Therapuatics. R.A. is a paid consultant for Bristol-Meyer Squibb, Merck, Astrazeneca. The remaining authors declare no other competing interests.

## Additional information

Thatcher Heumann[1,2,3], Carol Judkins[1,3], Keyu Li[4], Su Jin Lim[1,5], Jessica Hoare[1,3], Rose Parkinson[1,3], Haihui Cao[1,3], Tengyi Zhang[1,3,6], Jessica Gai[1,3,6], Betul Celiker[1,3], Qingfeng Zhu[1,3,6,7], Thomas McPhaul[1,6,8], Jennifer Durham[1,3], Katrina Purtell[1,3], Rachel Klein[1], Daniel Laheru[1,3,6], Ana De Jesus-Acosta[1,3,6], Dung T. Le[1,3,6], Amol Narang[1,6,9], Robert Anders[1,3,6,7], Richard Burkhart[1,3,6,8], William Burns[1,3,6,8], Kevin Soares[10], Christopher Wolfgang[11], Elizabeth Thompson[1,3,6,7], Elizabeth Jaffee[1,3,6,7], Hao Wang[1,3,5,12], Jin He[1,3,6,8,12] & Lei Zheng[1,3,6,8,12] ✉

[1]The Sidney Kimmel Comprehensive Cancer Center at Johns Hopkins, Department of Oncology, Cancer Convergence Institute and Bloomberg-Kimmel Institute for Cancer Immunotherapy, Johns Hopkins University School of Medicine, Baltimore, MD, USA. [2]Vanderbilt University Medical Center, Department of Hematology-Oncology, Nashville, TN, USA. [3]The Bloomberg-Kimmel Institute for Cancer Immunotherapy at Johns Hopkins, Baltimore, MD, USA. [4]Division of Pancreatic Surgery, Department of General Surgery, West China Hospital, Sichuan University, Chengdu, Sichuan, China. [5]Division of Quantitative Sciences, Department of Oncology, Johns Hopkins University School of Medicine, Baltimore, MD, USA. [6]The Pancreatic Cancer Precision Medicine Center of Excellence Program at Johns Hopkins, Baltimore, MD, USA. [7]Department of Pathology, Johns Hopkins University School of Medicine, Baltimore, MD, USA. [8]Department of Radiation Oncology, Johns Hopkins University School of Medicine, Baltimore, MD, USA. [9]Department of Surgery, Johns Hopkins University School of Medicine, Baltimore, MD, USA. [10]Department of Surgery, Memorial Sloan Kettering Cancer Center, New York, NY, USA. [11]Department of Surgery, New York University School of Medicine and NYU-Langone Medical Center, New York, NY, USA. [12]These authors jointly supervised this work: Hao Wang, Jin He, Lei Zheng. ✉e-mail: lzheng6@jhmi.edu

