## [Peer Review File · Nature Communications]

A Platform Trial of Neoadjuvant and Adjuvant Anti-Tumor Vaccination alone or in combination with PD-1 antagonist and CD137 agonist antibodies in Patients with Resectable Pancreatic AdenocarcinomaEditorial Note: Elements of this file have been redacted as they do not pertain to the peer review of this manuscript.

REVIEWER COMMENTS

Reviewer #1 (Remarks to the Author): with expertise in pancreatic cancer, immunotherapy

This manuscript reports on the results of a platform trial of various anti-tumor vaccine and PD-1 antagonist and CD137 agonist combinations in resectable pancreatic cancer. The authors are to be commended on this innovative trial design, and choosing the resectable setting to study the biological effects of the various treatment arms.

Overall the manuscript is well written and the tables and figures are easy to read and interpret.

There are 2 major limitations to this study related to the very small sample size and in-balance in adjuvant chemotherapy given between the arms, especially arm 3 vs 1 and 2.

1. In the abstract, and throughout the paper the efficacy results are overstated. The n of arm C is very small (10 patients) and the change in DFS is not statistically significant. A much higher proportion of patients in Arm C received FOLFIRINOX (70%) compared to approx 20% in arms A and B. Per the Conroy et al. adjuvant FOLFIRINOX trial (NEJM 2018), adjuvant FOLFIRINOX is significantly more efficacious than Gem. In this trial median DFS in the FOLFIRINOX arm was 21.6 months and median OS was 54.4 months. In the context of the very small numbers, Arm C in this trial had a better DFS (33.5 months) but significantly shorter OS (35.5 months). Furthermore the primary endpoint for all arms were stated to be biological/immunological endpoints. This should be emphasized and reported 1st in the results section in both abstract and main manuscript. In the abstract I would suggest stating that while a numerical difference was noted in DFS, it was not statistically different and there was in-balance in adjuvant chemotherapy given.

2. The introduction and methods section are well written.

3. In the results section, I would suggest moving immunologic endpoints ahead as efficacy as this was the primary endpoint of the trial.

4. In the efficacy results section I would suggest rewording and removing lines 432-439. It should be clearly stated that the difference between the arms was not statistically significant. It is very difficult to determine clinical significance with such a small sample size and the noted imbalance in adjuvant chemotherapy. The multivariate analysis noted in 437-439 is not statistically significant and I would suggest rewording or removing.

5. Lines 444-452 also overstate the OS differences. None of the analysis are statistically significant and it is very difficult to comment on trends in clinical significance with an n of 10. As noted in point 4 the multivariate analysis is again not-significant and overstates the trend.

6. In discussion lines 515-517, it states that arm C demonstrated promising and meaningful antitumor activity. I would consider rewording. Agree this is interesting and worthy of further study, but again the current wording seems to overstate the results given issues noted above. Also as noted above I would suggest discussing the biologic/immunologic endpoints 1st.

7. In lines 531-532 the authors should note that while the DFS in arms C compares favourably to adjuvant FOLFIRINOX, the median OS is much shorter- which may just be related to small sample size but should be noted.

8. In paragraph 558-575 the authors address the confounding related to imbalance of FOLFIRINOX. As noted above the hazard models were employed but again did not show a statistically significant difference between the arms. The other comparisons employed are very limited due to Arm C only

having 10 patients.

9. I do agree this combination is worthy of future study but I would suggest rewording the last sentence of this paragraph (573), as it again is overstating the efficacy results.

Overall this is very interesting study with novel design and interesting biological and immunological data. As noted in the above comments I think it would be significantly stronger if the immune/biological data was emphasized and the efficacy results were de-emphasized throughout. They are secondary endpoints, not adequately powered (with very small sample size) and limited further by significant heterogeneity in adjuvant chemo.

Reviewer #2 (Remarks to the Author): with expertise in pancreatic cancer, immunology

Heumann and colleagues report on the outcomes from a platform trial of GVAX +/- PD1 +/- TIGIT agonists in human PDAC patients. Overall, the authors are to be commended for the set up and executions of a potentially informative trial. And the merits of such a study are clear, if done well. The authors report on the safety of these combinations and initial efficacy in small cohorts. But primarily this is powered as a biomarker trials (aka not powered for efficacy). Despite these potential strengths the study at present is really limited by the parsimonious amount of biomarker data presented and new biologic findings. I think this is possibly addressable but would take some work; but would make the article impactful/more impactful.

Major comments.

1. The biomarker piece here is underwhelming and could use improvement.

a. The author quantitate CD3+ CD8+CD137+GRZB+ cells. The authors should show this data in more than one way. Do CD8 T cell numbers change (% of total) across the treatment. Does GRZB+ or CD137+ cells change among the T cells. AKA do more T cell become CD137+ or are there absolute number of CD137+ T cells climb with T cells.

b. Did the authors not measure anything else? This is a very limited analysis of T cells, and negligible for the TME. And the only parameter that appears to change is CD137 expression?

c. Do any of their biomarkers associate or show a trend with OS or PFS in Arm C (small numbers may make this hard, but the data are there).

d. Figure panel F does not specify treatment. I believe the only treatment with CD137+ T cells is full combination, so correlation here does not add much.

e. The authors say TLA counting in Fig4 legend, but appear to be counting cells not structures, this distinction needs clarity and/or data/methodology.

f. The authors should do more to show biologic impact of these therapies on T cells or other to enhance the impact of the study. In my opinion, as the clinical data lack statistical power for efficacy (which is ok), other than safety, this is the meat of the study. At current we learn TIGIT goes up, when you treat with TIGIT. I'd love to see them push this further.

2. In Supplemental Figure 4, the % of CD137+ T cells appears to split patients treated with any GVAX therapy. Which is interesting. But according to Figure 4C, there is only 1 or 2 patients outside of Arm C that have CD137+ T cells. Is this just powered by ArmC, which is almost statistical. Maybe a comparison to total CD8 T cells or other would give this more impact.

3. For biomarkers, it appears 10-14 patients make it to surgery (Figure 2), but the biomarker appears only to have 8/group. Why was this?

3. The authors should tailor their language closer to what the statistical data support. Especially in the abstract and results. The discussion can be more interpretation. There are a few examples but this one is the most important.

1. e.g. line 444 Patient treated with the full combination did not have a statistical difference in OS or PFS, yet the authors said "showed a clinically meaningful improvement in OS compared to....". $p=0.377$ and $p=0.279$. Maybe state the data, then add a line saying a trend

2. In the abstract, "combination Cy-GVAX+nivolumab+urelumab demonstrated a clinically- meaningful

improved DFS compared to Cy-GVAX alone" "(HR 0.55[95%CI 0.21,149],p=0.242)"
I formally don't disagree looks intriguing. But these statements are not statistically supported.

Minor Comments.

1. The authors should consider, where possible, simplify or clarify the table in figure 2. It is redundant with some of the text and a bit burdensome. (This is minor)
2. Figure 3 could use some improvement in layout. Y axis are very far from DFS/OS curves.
3. The images in Fig 4d-e are not the best.

Reviewer #3 (Remarks to the Author): with expertise in pancreatic cancer, immunotherapy

Neoadjuvant trials with short-course, upfront immunotherapy are advantageous for looking at tissue endpoints in resected specimens, and it removes the confounding effects of cytotoxic chemotherapy, so in that way the design is seen as a strength.

While the concept of platform trials is interesting, it is not novel and has been used and published in other cancer types.

GVAX has been around for a long time and has not shown to improve clinically meaningful outcomes, even with the addition of nivolumab.

While the results of the tissue analysis are interesting, the survival results of the triplet therapy are not statistically significant, and the patient numbers are too small to make any meaningful efficacy conclusions.

Post-op adjuvant chemo +/- XRT was not standardized, a potential confounder of the efficacy results.

What was the rationale for the "extended treatment Phase with nivolumab and does this amended change during the protocol further confound the results?"

30% of the patients in ARM C had T1 tumors (compared with 14-18% in A and B) which alone could account for the survival advantage seen in patients in the triplet therapy ARM.

Reviewer #4 (Remarks to the Author): with expertise in biostatistics, clinical trial study design

In this paper, the authors conducted a three-arms platform trial to demonstrate the feasibility of testing novel immunotherapy combinations in patients with resectable PDA. The primary endpoints were survival outcomes and immune endpoints. I have the following questions regarding to the statistical design and analysis:

1. Due to the small sample size, it is hard to tell that the HR for survival outcome demonstrate real clinical benefit. None of the p-values for HR is significant (<0.05), and the 95% upper quantiles for most HR are far beyond 1. More patients are needed to confirm the finding.
2. The description of sample size consideration is very unclear. The pre-specified effect size is huge, do you have any data to support such setting? What test do you used for power calculation? Do you consider the multiple comparison for sample size?
3. The endpoints are inconsistent across different arms. "The primary endpoint for Arms A and B was IL17A expression in vaccine-induced lymphoid aggregates in resected PDAs from patients treated with the combination of Cy-GVAX with or without nivolumab (19). The primary biologic endpoint for Arm C was CD8+CD137+T cell density within tumor regions of interest (containing at least one TLA) in surgically resected specimens." This should cause problems for statistical testing between Arm A/B and Arm C.
4. Why the non-parametric analysis is used for immune outcome, not the t-test? Do you check the normality of the data? Better provide the Q-Q plot.

5. Page 181, “Cy-GVAX alone (HR 0.55[95%CI 0.21,149],p=0.242)”, should 149 be 1.49? Also, why the results (p-values) for the immune endpoints are not mentioned here?

RESPONSES TO REVIEWER COMMENTS

Reviewer #1 (Remarks to the Author): with expertise in pancreatic cancer, immunotherapy

This manuscript reports on the results of a platform trial of various anti-tumor vaccine and PD-1 antagonist and CD137 agonist combinations in resectable pancreatic cancer. The authors are to be commended on this innovative trial design, and choosing the resectable setting to study the biological effects of the various treatment arms.

Overall the manuscript is well written and the tables and figures are easy to read and interpret.

Author reply: We very much appreciate this positive and enthusiastic feedback regarding our clinical trial design and manuscript.

There are 2 major limitations to this study related to the [1]very small sample size and [2]in-balance in adjuvant chemotherapy given between the arms, especially arm 3 vs 1 and 2.

Author reply: We agree with the reviewer that these are 2 major limitations. We have addressed these further in the author replies below and within the manuscript proper, as indicated.

1. In the abstract, and throughout the paper the efficacy results are overstated. The n of arm C is very small (10 patients) and the change in DFS is not statistically significant. A much higher proportion of patients in Arm C received FOLFIRINOX (70%) compared to approx 20% in arms A and B. Per the Conroy et al. adjuvant FOLFIRINOX trial (NEJM 2018), adjuvant FOLFIRINOX is significantly more efficacious than Gem. In this trial median DFS in the FOLFIRINOX arm was 21.6 months and median OS was 54.4 months. In the context of the very small numbers, Arm C in this trial had a better DFS (33.5 months) but significantly shorter OS (35.5 months). Furthermore the primary endpoint for all arms were stated to be biological/immunological endpoints. This should be emphasized and reported 1st in the results section in both abstract and main manuscript. In the abstract I would suggest stating that while a numerical difference was noted in DFS, it was not statistically different and there was in-balance in adjuvant chemotherapy given.

Author reply: Thank you for the feedback. We have made the following revisions to address this comment:

Original	Revised
Results: Forty patients (n=16[A],n=14[B],n=10[C]) were eligible for efficacy analysis. Median DFS(95% CI) was 13.90mo(5.59,NR), 14.98mo(7.95,44.09) and 33.51(16.76,NR) for Arms A/B/C, respectively. Combination Cy-GVAX+nivolumab+urelumab demonstrated a clinically-meaningful improved DFS compared to Cy-GVAX alone (HR	Results: Forty patients (n=16[A],n=14[B],n=10[C]) were eligible for efficacy analysis. Treatment with combination Cy-GVAX+nivolumab+urelumab met the primary endpoint by significantly increasing intratumoral CD8+CD137+ and CD8+CD137+GZMB+ T cells compared to Cy-GVAX± Nivolumab treatment. Median DFS(95% CI) was 13.90mo(5.59,NR),

0.55[95%CI 0.21,1.49],p=0.242) and Cy-GVAX+nivolumab (HR 0.51[95%CI 0.19,1.35],p=0.173). All three treatments were well tolerated. Treatment with combination Cy-GVAX+nivolumab+urelumab met the primary endpoint by significantly increasing intratumoral CD8+CD137+ and CD8+CD137+GZMB+ T cells compared to Cy-GVAX± Nivolumab treatment.	14.98mo(7.95,44.09) and 33.51(16.76,NR) for Arms A,B,C, respectively. While the combination Cy-GVAX+nivolumab+urelumab demonstrated numerically-improved DFS compared to Cy-GVAX alone (HR 0.55[95%CI 0.21,1.49],p=0.242) and Cy-GVAX+nivolumab (HR 0.51[95%CI 0.19,1.35],p=0.173), this was not statistically significant and was limited by a small sample size as well as imbalance in standard of care adjuvant chemotherapy regimens. All three treatments were well tolerated.
--	---

2. The introduction and methods section are well written.

Author reply: We appreciate this feedback.

3. In the results section, I would suggest moving immunologic endpoints ahead as efficacy as this was the primary endpoint of the trial.

Author reply: Thank you for the feedback. This has been restructured to reflect the above recommendation.

4. In the efficacy results section I would suggest rewording are removing lines 432-439. It should be clearly stated that the difference between the arms was not statistically significant. It is very difficult do determine clinical significance with such a small sample size and the noted imbalance in adjuvant chemotherapy. The multivariate analysis noted in 437-439 is not statistically significant and I would suggest rewording or removing.

Author reply: Thank you for the feedback. We have made the following revisions to address this comment:

Original	Revised
Efficacy: At median follow up times of 23.1 [Arm A], 26.1 [Arm B], and 31.6 [Arm C] months (mo), median DFS (95% CI) was 13.90 mo (5.59, NR), 14.98 mo (7.95, 44.09) and 33.51mo (16.76, NR) for Arms A, B, C, respectively (Table 2, Fig 3). Compared to Cy-GVAX alone (Arm A), adding nivolumab to Cy-GVAX (Arm B) did not improve DFS (HR 1.09 [95% CI 0.50, 2.40], p=0.829) (Table 2, Fig 3). Detecting true statistical	Efficacy: At median follow up times of 23.1 [Arm A], 26.1 [Arm B], and 31.6 [Arm C] months (mo), median DFS (95% CI) was 13.90 mo (5.59, NR), 14.98 mo (7.95, 44.09) and 33.51mo (16.76, NR) for Arms A, B, C, respectively (Table 2, Fig 3). Detecting true statistical significance was limited due to the small number of patients within each treatment arm. In context of this, compared to Cy-GVAX alone (Arm A), adding nivolumab

significance was limited due to the small number of patients within each treatment arm. However, despite this, patients treated with the combination of urelumab, nivolumab, and Cy-GVAX (Arm C) demonstrated a clinically-compelling benefit in DFS when compared against those treated with Cy-GVAX alone (HR 0.55 [95% CI 0.21, 1.49], p=0.242) or Cy-GVAX with nivolumab (HR 0.51 [95% CI 0.19, 1.35], p=0.173) (Table 2, Fig 3). This trend persisted after controlling for age, nodal spread, and adjuvant chemotherapy regimen (HR=0.64 [95% CI 0.19-2.19], p=0.478 compared with Arm A; HR=0.48 [95% CI 0.15-1.60], p=0.232 compared with Arm B) (Table S1).	to Cy-GVAX (Arm B) did not improve DFS (HR 1.09 [95% CI 0.50, 2.40], p=0.829) (Table 2, Fig 3). Patients treated with the combination of urelumab, nivolumab, and Cy-GVAX (Arm C) demonstrated numerically-improved DFS when compared against those treated with Cy-GVAX alone (HR 0.55 [95% CI 0.21, 1.49], p=0.242) or Cy-GVAX with nivolumab (HR 0.51 [95% CI 0.19, 1.35], p=0.173) (Table 2, Fig 3), but did not reach statistical significance. This favorable HR trend, though again not statistically significant, persisted after controlling for age, nodal spread, and adjuvant chemotherapy regimen (HR=0.64 [95% CI 0.19-2.19], p=0.478 compared with Arm A; HR=0.48 [95% CI 0.15-1.60], p=0.232 compared with Arm B) (Table S1).
---	--

5. Lines 444-452 also overstate the OS differences. None of the analysis are statistically significant and it is very difficult to comment on trends in clinical significance with an n of 10. As noted in point 4 the multivariate analysis is again not-significant and overstates the trend.

Author reply: Thank you for the feedback. We have made the following revisions to address this comment:

Original	Revised
Median OS (95% CI) was 23.59 mo (13.27, NR), 27.01 mo (20.76, NR), and 35.55 mo (17.74, NR) for Arms A, B, C, respectively (Table 2, Fig 3). Compared to Cy-GVAX alone, adding PD1 to Cy-GVAX did not improve OS (HR=1.11 [95% CI 0.47, 2.63], p=0.813) (Table 2, Fig 3). Patients treated with the combination of CD137 + PD1 + Cy-GVAX showed a clinically meaningful improvement in OS when compared against those treated with Cy-GVAX alone (HR 0.59 [95% CI 0.18, 1.91], p=0.377) and in combination PD1 (HR=0.53 [95% CI 0.17, 1.67], p=0.279) (Table 2, Fig 3). Similar to DFS, this clinically-meaningful HR persisted after controlling for age, nodal spread, and adjuvant chemotherapy regimen (HR=0.41 [95% CI 0.10-1.62], p=0.202 compared to Arm A; HR=0.59 (95% CI 0.18-1.91),	Median OS (95% CI) was 23.59 mo (13.27, NR), 27.01 mo (20.76, NR), and 35.55 mo (17.74, NR) for Arms A, B, C, respectively (Table 2, Fig 3). Compared to Cy-GVAX alone, adding PD1 to Cy-GVAX did not improve OS (HR=1.11 [95% CI 0.47, 2.63], p=0.813) (Table 2, Fig 3). Patients treated with the combination of CD137 + PD1 + Cy-GVAX showed a numerically-improved OS when compared against those treated with Cy-GVAX alone (HR 0.59 [95% CI 0.18, 1.91], p=0.377) and in combination PD1 (HR=0.53 [95% CI 0.17, 1.67], p=0.279) (Table 2, Fig 3), but did not reach statistical significance. Similar to DFS, this favorable HR persisted after controlling for age, nodal spread, and adjuvant chemotherapy regimen (HR=0.75 [95% CI 0.18-3.10], p=0.0692 compared to Arm A; HR=0.41 (95% CI 0.10-

p=0.377 compared to Arm B) (Table 2). On MVA, presence of nodal spread at time of surgery correlated with worse OS (HR=2.92 [1.02-8.32], p=0.045) and trended towards worse DFS (HR=2.21 [0.88-5.53], p=0.091) (Table S1, Table S2). Type of SOC adjuvant systemic treatment was not significantly correlated with DFS or OS in our study sample (Table S1, Table S2, Fig S3).	1.62), p=0.202 compared to Arm B) (Table 2), but did not reach statistical significance. On MVA, presence of nodal spread at time of surgery correlated with worse OS (HR=2.92 [1.02-8.32], p=0.045) and trended towards worse DFS (HR=2.21 [0.88-5.53], p=0.091) (Table S1, Table S2). Type of SOC adjuvant systemic treatment was not significantly correlated with DFS or OS in our study sample nor was tumor-stage (Table S1, Table S2, Fig S3).
---	--

6. In discussion lines 515-517, it states that arm C demonstrated promising and meaningful antitumor activity. I would consider rewording. Agree this is interesting and worthy of further study, but again the current wording seems to overstate the results given issues noted above. Also as noted above I would suggest discussing the biologic/immunologic endpoints 1st.

Author reply: Thank you for the feedback. We have made the following revisions to address this comment:

Original	Revised
This triplet combination demonstrated promising and meaningful antitumor activity that may enhance DFS in resected PDA patients treated in the perioperative and post-adjuvant settings. In addition, this triplet regimen significantly increased percentages of tumor-infiltrating activated T cells (CD3+CD8+CD137+ T cells) and activated, cytotoxic effector T cells (CD3+CD8+GZMB+CD137+ T cells), meeting its the primary endpoint. This study also suggests that increasing the number of infiltrating effector T cell by itself may not be sufficient and that further optimization of effector T cell quality and activation, such as with an immune agonist mAb, may help enhance antitumor immune response to immunotherapy in PDA.	The triplet combination met its primary endpoint: demonstrating promising tumor microenvironment changes by significantly increasing the percentages of tumor-infiltrating activated T cells (CD3+CD8+CD137+ T cells) and activated, cytotoxic effector T cells (CD3+CD8+GZMB+CD137+ T cells). The observed treatment-related changes suggest that increasing the number of infiltrating effector T cells by itself may not be sufficient and that further optimization of effector T cell quality and activation, such as with an immune agonist mAb, may help enhance antitumor immune response to immunotherapy in PDA. Acknowledging that clinical outcomes as secondary endpoints were limited by small size and imbalance in standard adjuvant chemotherapy regimens, the triplet regimen did demonstrate numerically-improved DFS in resected PDA patients. While this did not reach statistical significance, it merits further exploration for

	use in perioperative and post-adjuvant settings.
--	--

7. In lines 531-532 the authors should note that while the DFS in arms C compares favourably to adjuvant FOLFIRINOX, the median OS is much shorter- which may just be related to small sample size but should be noted.

Author reply: Thank you for the feedback. We have made the following revisions to address this comment:

Original	Revised
Patients randomized to Cy-GVAX alone or in combination with nivolumab, experienced mDFS and mOS intervals similar to those results established in phase III trials of their respective SOC adjuvant chemotherapy regimen (23-25) and were consistent with our previous trials of Cy-GVAX in the resectable PDA patient population (8, 20) (Table S4). In context of our sample size, we did observe a strong trend toward DFS benefit in patients treated with the triple combination of Cy-GVAX, Nivolumab, & Urelumab. This combined with the associated treatment-related increases tumor-infiltrating activated effector T cells demonstrates a potential efficacy signal for this novel IO combination that, when placed in the context of previous adjuvant IO and landmark phase III chemotherapy trials in resectable PDA patients (Table S4), should merit further study.	Patients randomized to Cy-GVAX alone or in combination with nivolumab, experienced mDFS and mOS intervals similar to those results established in phase III trials of their respective SOC adjuvant chemotherapy regimen (23-25) and were consistent with our previous trials of Cy-GVAX in the resectable PDA patient population (8, 20) (Table S4). In context of a small sample size and imbalance in standard adjuvant therapy, we did observe a numerically, but non-statically significant, improvement in DFS for patients treated with the triple combination of Cy-GVAX, Nivolumab, & Urelumab. This combined with the associated treatment-related increases tumor-infiltrating activated effector T cells, may be a potential efficacy signal for this novel IO combination that, when placed in the context of DFS outcomes in previous adjuvant IO and landmark phase III chemotherapy trials in resectable PDA patients (Table S4), should merit further study. While the mOS comparisons to these same appeared less favorable (e.g. PRODIGE mOS 54.4 mo), it should be noted that conclusions about mOS may be of limited value given the small patient numbers, need for further follow up time to allow OS outcomes to mature, and, most significantly, influence of salvage therapy/subsequent lines of treatment rather than the study intervention. Because of this, DFS was favored as the more appropriate endpoint to

	evaluate the impact of this study intervention in this treatment setting.
--	---

8. In paragraph 558-575 the authors address the confounding related to imbalance of FOLFIRINOX. As noted above the hazard models were employed but again did not show a statistically significant difference between the arms. The other comparisons employed are very limited due to Arm C only having 10 patients.

Author reply: Thank you for the feedback. We agree that the small sample size in Arm C has limited the conclusions that could be drawn from this study; however, this study has provided the clinical efficacy and immune response signals rapidly with only a small number of patients tested. We are currently planning a randomized phase 2 study designed (and powered) to assess the clinical efficacy of the triple combination in Arm C. We have made the following revisions to address this comment:

Original	Revised
To address a potential confounder, the higher percentage of Arm C patients receiving adjuvant (m)FOLFIRINOX compared to patients in Arms A & B, multiple strategies were employed to evaluate the additive contribution of IO triplet combination to the observed DFS trends. The survival hazard models attempted to control for chemo regimens (Table S1, Table S2). Additionally, across all treatment arms, patients who received adjuvant (m)FOLFIRINOX appeared to have similar DFS when compared to the collective study participants who were treated with gemcitabine-based regimens (Fig S3). Finally, Arm C patients were compared against a historical control cohort of resected PDA treated at Johns Hopkins Sidney Kimmel Cancer Center during the time of Arm C’s enrollment. When matched 3:1 on adjuvant chemo regimen, age, and nodal disease status with propensity score matching (Table S5, Fig S7), Arm C patients maintained a favorable DFS HR: Arm C mDFS = 33.02 mo; Historical Control mDFS= 20.83 mo; stratified HR 0.72 [0.29-1.80], p=0.480 (*DFS was measured starting the day of surgery for both groups) (Table S6, Fig S7). It should be noted that the historical cohort’s DFS carries a potential lead-time bias due to the follow up and restaging scan	It is important to acknowledge the higher percentage of Arm C patients receiving adjuvant (m)FOLFIRINOX compared to patients in Arms A and B. To address this potential confounder, multiple strategies were employed to evaluate the additive contribution of IO triplet combination to the observed DFS trends. The multivariable survival analysis attempted to control for chemo regimen. Additionally, across all treatment arms, patients who received adjuvant (m)FOLFIRINOX appeared to have similar DFS when compared to the collective study participants who were treated with gemcitabine-based regimens. Finally, Arm C patients were also compared against a matched-historical control cohort. There are clear limitations of this study and the above analyses driven largely by the sample size. However, the early clinical and immune response signals observed in this small cohort support a follow up, randomized, phase 2 study designed, and powered, to assess the clinical efficacy of the triple IO combination used in Arm C.

schedule being more stringent for patients on the trial. Even with the increased matching ratio, the sample size remains modest for comparison. However, the notable difference in median DFS, and visible separation on the survival curves, argue for a larger, follow up phase II trial powered for clinical outcomes with uniform IO dosing, and SOC adjuvant regimes.	
---	--

9. I do agree this combination is worthy of future study but I would suggest rewording the last sentence of this paragraph (573), as it again is overstating the efficacy results.

Author reply: We agree and have made revisions to address this comment. Please see response to Reviewer #1, comment #8.

Overall this is very interesting study with novel design and interesting biological and immunological data. As noted in the above comments I think it would be significantly stronger if the immune/biological data was emphasized and the efficacy results were de-emphasized throughout. They are secondary endpoints, not adequately powered (with very small sample size) and limited further by significant heterogeneity in adjuvant chemo.

Author reply: We appreciate Reviewer's comments and constructive suggestions. We agree and hope the above revisions are satisfactory.

Reviewer #2 (Remarks to the Author): with expertise in pancreatic cancer, immunology

Heumann and colleagues report on the outcomes from a platform trial of GVAX +/- PD1 +/- TIGIT agonists in human PDAC patients. Overall, the authors are to be commended for the set up and executions of a potentially informative trial. And the merits of such a study are clear, if done well. The authors report on the safety of these combinations and initial efficacy in small cohorts. But primarily this is powered as a biomarker trials (aka not powered for efficacy). Despite these potential strengths the study at present is really limited by the parsimonious amount of biomarker data presented and new biologic findings. I think this is possibly addressable but would take some work; but would make the article impactful/more impactful.

Major comments.

1. The biomarker piece here is underwhelming and could use improvement.

Author reply: We appreciate Reviewer's interest in our biomarker data. As our study is a platform design, the correlative studies from Arm A and Arm B were conducted before Arm C was proposed. These correlative studies have recently been published in Li et al. Cancer Cell

2022. We have more recently conducted the biomarker study on the specimens from Arm C, with a focus on the single cell RNA sequencing analysis for hypothesis generation. We found it is difficult to combine them all into one manuscript. In this manuscript, we focus on the primary (immune efficacy) and secondary (clinical efficacy) endpoints and include more hypothesis testing work, but not hypothesis generating work.

In the resubmitted manuscript, we include a manuscript in preparation with the single cell RNA sequencing analysis as Supplemental Data for Reviewers Only. [Editorial Note: redacted]

We have also included additional multiplex immunohistochemistry data on more immune subtypes infiltrating the tumors as a supplemental figures for this manuscript (new supplemental **Figure S9 and S10**). See below:

Fig S9: Comparison of the Densities of Multiple Immune Cell Subtypes between Treatment Arms. [a] CD45+CD3+CD8+ T cells; [b] CD45+CD3+CD8+PD-1+ T cells; [c] CD45+CD3+CD4+ T cells; [d] CD45+CD3+CD4+Foxp3 T cells; [e] CD45+CD3-CSF-1R+CD68+CD163- M1-like macrophages; [f] CD45+CD3-CSF-1R+CD68+CD163+ M2-like macrophages; [g] CD45+CD3-CD66b+ neutrophils. Arm A: n=9; Arm B: n=10; Arm C: n=8. Treatment arms as indicated. Wilcoxon tests were performed; p values were shown: *<0.05; **<0.01; ***<0.001; if not shown, non-significance.

Fig S10: Potential Effects of CD137 Agonist Treatment on T Cell Exhaustion, Activation and Trafficking. [a] Samples in Arm C were subgrouped, according to the density of CD137+CD8+ T cells in TLAs by using the mean of the density as cutoff, into two cohorts: low vs. high CD137+ T cells. The density of CD45+CD3+CD8+TIGIT+ T cells was compared between the two cohorts. [b] The percentage of CD45+CD3+CD8+CD137+ T cells among CD45+CD3+CD8+ T cells was compared between treatment arm; [c] The percentage of CD45+CD3+CD8+GZMB+ T cells among CD45+CD3+CD8+ T cells was compared between treatment arm; [d] The density of CD45+CD3+CD8+CD137+ T cells in the tumor vicinity area outside TLAs, calculated as the percentage among all cells, was compared between treatment arms. Arm A: n=9; Arm B: n=10; Arm C: n=8. Treatment arms as indicated. Wilcoxon tests were performed; p values were shown: * < 0.05; ** < 0.01; *** < 0.001; if not shown, non-significance.

2. The author quantitate CD3+ CD8+CD137+GRZB+ cells. The authors should show this data in more than one way. Do CD8 T cell numbers change (% of total) across the treatment. Does GRZB+ or CD137+ cells change among the T cells. AKA do more T cell become CD137+ or are there absolute number of CD137+ T cells climb with T cells.

Author reply: Thank you for the comment. We have added the results as suggested by Reviewer in the new supplemental figures (**Figure S9 and S10**) as above. We chose to quantitate CD3+ CD8+CD137+GRZB+ cells according to the correlative studies from Arm A and Arm B (Li et

al. Cancer Cell 2022). As anticipated, CD8+ T cell density does not significantly change across the treatment arms. We calculated the density of immune cell subtypes in the endpoint analysis as the percentage among All Cells to be consistent with our prior publication. However, as shown in **Figure S10**, *the percentage of CD137+CD8+ T cells, but not that GZMB+CD8+ T cells, among CD8+ T cells significantly increased in Arm C, suggesting that a subset of CD8+ T cells, which is likely a subset of GZMB+ cytotoxic T cells considering their strong correlation with CD137+CD8+ T cells (Fig S3), was converted to activated effector T cells following CD137 agonist treatment (Fig S10b,c).*

3. Did the authors not measure anything else? This is a very limited analysis of T cells, and negligible for the TME. And the only parameter that appears to change is CD137 expression?

Author reply: Thank you for the comment. As mentioned in our reply to Comment #1, our study is a platform design and the correlative studies from Arm A and Arm B were conducted before Arm C was proposed. These correlative studies have recently been published in Li et al. Cancer Cell 2022. We have more recently conducted the biomarker study on the specimens from Arm C, with a focus on the single cell RNA sequencing analysis for hypothesis generation. We found it is difficult to combine them all into one manuscript. In this manuscript, we focus on the primary (immune efficacy) and secondary (clinical efficacy) endpoints and include more hypothesis testing work, but not hypothesis generating work.

In the resubmitted manuscript, we include a manuscript in preparation with the single cell RNA sequencing analysis as Supplemental Data for Reviewers Only. [Editorial Note: redacted]

We have also included additional multiplex immunohistochemistry data on more immune subtypes infiltrating the tumors as a supplemental figures for this manuscript (**Figure S9 and S10**):

The general CD8+ T cells increased in Arm B, but did not further increase in Arm C. Interestingly, although PD-1+CD8+ T cells decreased in Arm B compared to Arm A as previously reported, PD-1+CD8+ T cells modestly increased in Arm C compared to Arm B likely as a result of T cell activation by CD137 agonist. More interestingly, Foxp3+CD4+ Tregs significantly increase in Arm C compared to Arms A and B, consistent with the role of CD137 in Tregs as previously suggested. Whether this induction of Treg would suppress antitumor immune response remains to be investigated. Analysis of myeloid cell subtypes showed that CD137 agonist decreased both M1 and M2-like tumor-associated macrophages, but did not change tumor-associated neutrophils significantly.

4. Do any of their biomarkers associate or show a trend with OS or PFS in Arm C (small numbers may make this hard, but the data are there).

Author reply: We have performed the analysis to correlate the biomarkers in surgical tissues with DFS and OS, including CD8+CD137+ T cells, CD8+GZMB+ T cells and CD8+CD137+GZMB+ T cells, and the results are summarized in **Table S1** (below) and **Figures S2, S4, and S5**. The data seem to suggest that the higher densities of these cells of interest, particularly in the case of CD8+CD137+ T cells and CD8+CD137+GZMB+ T cells, are

associated (or trend towards) improved DFS/OS outcomes. Among 10 patients in Arm C, only 2 patients would be considered to have a shorter survival by using the same cutoff as we previously used for Arm A and Arm B (Li et al. *Cancer Cell* 2022); therefore, we could not meaningfully interpret whether these respective cell densities correlated with clinical efficacy outcomes.

Table S1: Cox Proportional Hazard Model (Univariate) for Disease-Free and Overall Survival Tumor Tissue Covariates

	Frequency (%)	DFS		OS	
		HR (95%CI)	p	HR (95%CI)	p
CD3+CD8+CD137+ T cells Density^{ab}					
$\leq 0.41\%^d$	12 (52.2)	Ref	-	Ref	-
$> 0.41\%^d$	11 (47.8)	0.30 (0.11-0.86)	0.026	0.61 (0.22-1.70)	0.349
CD3+CD8+GZMB+ T cells Density^{bc}					
$\leq 2.1\%^d$	14 (50.0)	Ref	-	Ref	-
$> 2.1\%^d$	14 (50.0)	0.62 (0.26-1.50)	0.291	1.12 (0.45-2.76)	0.813
CD3+CD8+CD137+GZMB+ T cells Density^{ab}					
$\leq 0.01\%^d$	13 (56.5)	Ref		Ref	
$> 0.01\%^d$	10 (43.5)	0.41 (0.14-1.17)	0.095	0.41(0.13-1.29)	0.127

[a] The multiplex immunohistochemistry (mIHC) workflow was able to be performed on surgical specimens from 23 study patient: n=7 (Arm A), n=8(Arm B), n= 8 (Arm C); [b] Reflects an averaged proportion of T cell subtype within evaluated ROIs/TLAs per specimen. This was chosen instead of absolute numbers to reflect the proportion of this cell type of interest and to normalize comparisons between ROIs/TLAs within and across resected samples; [c] The multiplex immunohistochemistry (mIHC) workflow was able to be performed on surgical specimens from 28 study patient: n=10 (Arm A), n=10 (Arm B), n= 8 (Arm C); [d] Grouped by median across groups (above and below).

5. Figure panel F does not specify treatment. I believe the only treatment with CD137+ T cells is full combination, so correlation here does not add much.

Author reply: We agree with the comment and have moved panel F to supplemental data (now Fig S3).

6. The authors say TLA counting in Fig4 legend, but appear to be counting cells not structures, this distinction needs clarity and/or data/methodology.

Author reply: We apologize the confusion. The figure indeed shows the cell counting, not TLA counting. We have revised the figure legend.

Original	Revision
Fig 4: Combination GVAX, Nivolumab, and Urelumab Increases Infiltrating CD3+CD8+CD137+ and CD3+CD8+CD137+GZMB+ T Cells. [a] Scatter Plot Showing Proportion of CD3+CD8+CD137+ Infiltrating T cells in Tertiary Lymphoid Aggregates (averaged per specimen/ID) by Treatment Arm [n=23]; [b] Scatter Plot Showing Proportion of CD3+CD8+GZMB+ Infiltrating T cells in Tertiary Lymphoid Aggregates (averaged per specimen/ID) by Treatment Arm [n=28]; [c] Scatter Plot Showing Proportion of CD3+CD8+CD137+GZMB+ Infiltrating T cells in Tertiary Lymphoid Aggregates (averaged per specimen/ID) by Treatment Arm [n=23]	Fig 3: Combination GVAX, Nivolumab, and Urelumab Increases Infiltrating CD3+CD8+CD137+ and CD3+CD8+CD137+GZMB+ T Cells. [a] Shown was one representative ROI that contains TLA and epithelial neoplastic cells in the vicinity; quantification was done within TLA and the tumor vicinity area outside TLA, respectively. [b] Comparison of the density of CD3+CD8+CD137+ T cells within the TLA among treatment arms as indicated [n=23]; [c] Comparison of the density of CD3+CD8+GZMB+ T cells within TLA among treatment arms as indicated [n=28]; [d] Comparison of the density of CD3+CD8+CD137+GZMB+ T cells within TLA among treatment arms as indicated [n=23]. Wilcoxon tests were performed and Bonferroni corrected p values were shown: *<0.05; **<0.01; ns, non-significance. [e] Representative co-registered images of multiplex IHC showing CD3+CD8+CD137+ T cells within a tumor ROI; [f] Representative co-registered images of multiplex IHC showing CD3+CD8+GZMB+ T cells within a tumor ROI. In a,e,f, pseudocolors assigned to each marker as indicated.

7. The authors should do more to show biologic impact of these therapies on T cells or other to enhance the impact of the study. In my opinion, as the clinical data lack statistical power for efficacy (which is ok), other than safety, this is the meat of the study. At current we learn TIGIT goes up, when you treat with TIGIT. I'd love to see them push this further.

Author reply: We appreciate the comment. As mentioned above, in the resubmitted manuscript, we include a manuscript in preparation with the single cell RNA sequencing analysis as the Supplemental Data for Reviewers Only. [Editorial Note: redacted]

We also selected a few results of multiplex immunohistochemistry (new supplemental **Figure S9 and S10 above**) on more immune subtypes infiltrating the tumors as a supplemental figure for this manuscript. In the new supplemental Figure S10 below, we included the analysis of TIGIT+ T cells:

We also examined TIGIT+CD8+ T cells in TLAs in post-neoadjuvant immunotherapy tumors in Arm C (Fig S10a) and found that higher density of CD137+ T cells in TLAs is associated in a trend with lower density of TIGIT+CD8+ T cells, supporting our previously developed hypothesis that CD137 agonist treatment may overcome the T cell exhaustion(19). As we only recently developed the TIGIT multiplex IHC, we only stained TIGIT on the specimens from Arm C.

Fig S10: Potential Effects of CD137 Agonist Treatment on T Cell Exhaustion, Activation and Trafficking. [a] Samples in Arm C were subgrouped, according to the density of CD137+CD8+ T cells in TLAs by using the mean of the density as cutoff, into two cohorts: low vs. high CD137+ T cells. The density of CD45+CD3+CD8+TIGIT+ T cells was compared between the two cohorts. [b] The percentage of CD45+CD3+CD8+CD137+ T cells among CD45+CD3+CD8+ T cells was compared between treatment arm; [c] The percentage of CD45+CD3+CD8+GZMB+ T cells among CD45+CD3+CD8+ T cells was compared between treatment arm; [d] The density of CD45+CD3+CD8+CD137+ T cells in the tumor vicinity area outside TLAs, calculated as the percentage among all cells, was compared between treatment arms. Arm A: n=9; Arm B: n=10; Arm C: n=8. Treatment arms as indicated. Wilcoxon tests were performed; p values were shown: *<0.05; **<0.01; ***<0.001; if not shown, non-significance.

8. In Supplemental Figure 4, the % of CD137+ T cells appears to split patients treated with any GVAX therapy. Which is interesting. But according to Figure 4C, there is only 1 or 2 patients

outside of Arm C that have CD137+ T cells. Is this just powered by Arm C, which is almost statistical. Maybe a comparison to total CD8 T cells or other would give this more impact.

Author reply: The reviewer has highlighted a very important point. The percentage of CD137+ T cells is significantly different between Arm C and Arm A/B although this is the primary endpoint to be tested and thus is anticipated. Nevertheless, tumors in Arm A and Arm B also have CD137+ T cells although their numbers are very low and therefore appeared to be “zero”. In Li et al. Cancer Cell 2022, we studied CD137+ cells in Arm A and Arm B, whose density, albeit low, still positively correlated with outcome in Arm B, leading to the design of Arm C in our platform trial by using anti-CD137 agonist antibody to sustain the activation of CD137+ T cells. We have included more results (**Figure S9 and S10**) and also expanded the analysis below as suggested by Reviewer:

Although the general CD8+ T cells in TLAs did not increase in Arm C compared to Arm B (Fig S9a), the percentage of CD137+CD8+ T cells, but not that GZMB+CD8+ T cells, among CD8+ T cells significantly increased in Arm C, suggesting that a subset of CD8+ T cells, which is likely a subset of GZMB+ cytotoxic T cells considering their strong correlation with CD137+CD8+ T cells (Fig S3), was converted to activated effector T cells following CD137 agonist treatment (Fig S10b,c). As previously reported (19), CD8+CD137+ T cells were essentially restricted in TLAs with minimal-to-no CD8+CD137+ T cells seen in the tumor vicinity outside TLAs in Arms A and B. In contrast, this activated T cell subtype made up 2-4% cells in the tumor vicinity outside TLAs within the same ROIs in PDAs from Arm C (Fig 3e; Fig S10d), suggesting that activated T cells may have migrated from TLAs to the vicinity of neoplastic cells.

Fig S10: Potential Effects of CD137 Agonist Treatment on T Cell Exhaustion, Activation and Trafficking. [a] Samples in Arm C were subgrouped, according to the density of CD137+CD8+ T cells in TLAs by using the mean of the density as cutoff, into two cohorts: low vs. high CD137+ T cells. The density of CD45+CD3+CD8+TIGIT+ T cells was compared between the two cohorts. [b] The percentage of CD45+CD3+CD8+CD137+ T cells among CD45+CD3+CD8+ T cells was compared between treatment arm; [c] The percentage of CD45+CD3+CD8+GZMB+ T cells among CD45+CD3+CD8+ T cells was compared between treatment arm; [d] The density of CD45+CD3+CD8+CD137+ T cells in the tumor vicinity area outside TLAs, calculated as the percentage among all cells, was compared between treatment arms. Arm A: n=9; Arm B: n=10; Arm C: n=8. Treatment arms as indicated. Wilcoxon tests were performed; p values were shown: *<0.05; **<0.01; ***<0.001; if not shown, non-significance.

9. For biomarkers, it appears 10-14 patients make it to surgery (Figure 2), but the biomarker appears only to have 8/group. Why was this?

Author reply: Thank you for pointing this out. This study has pre-defined the criteria for immune endpoint analysis by focusing on the vaccine-induced TLAs and also maintaining a continuity of the endpoint analysis for Arm A and Arm B as described in Li et al. Cancer Cell 2022. *Three rectangle regions of interest (ROIs) of approximately 3000*3000 pixels each containing one tertiary lymphoid aggregate (TLA) and epithelial neoplastic cells in the vicinity, known to be representative of the larger whole tumor based on prior study, were chosen for analysis. This has been described in the Methods. We further clarified that tumors without an identifiable ROI that contained epithelial neoplastic cells in the vicinity of TLAs were excluded from the analysis. The cases were excluded from the analysis because no tumor cells were identified in the vicinity of TLAs within the same ROI, likely due to the tumor pathologic response to the neoadjuvant triple immune combo treatment.*

10. The authors should tailor their language closer to what the statistical data support. Especially in the abstract and results. The discussion can be more interpretation. There are a few examples but this one is the most important.

1. e.g. line 444 Patient treated with the full combination did not have a statistical difference in OS or PFS, yet the authors said “showed a clinically meaningful improvement in OS compared to....”. p=0.377 and p=0.279. Maybe state the data, then add a line saying a trend

Author reply: Thank you for the feedback. We have made the following revisions to address this comment:

Original	Revised
Efficacy: At median follow up times of 23.1 [Arm A], 26.1 [Arm B], and 31.6 [Arm C] months (mo), median DFS (95% CI) was 13.90 mo (5.59, NR), 14.98 mo (7.95, 44.09) and 33.51mo (16.76, NR) for Arms A, B, C,	Efficacy: At median follow up times of 23.1 [Arm A], 26.1 [Arm B], and 31.6 [Arm C] months (mo), median DFS (95% CI) was 13.90 mo (5.59, NR), 14.98 mo (7.95, 44.09) and 33.51mo (16.76, NR) for Arms A, B, C,

respectively (Table 2, Fig 3). Compared to Cy-GVAX alone (Arm A), adding nivolumab to Cy-GVAX (Arm B) did not improve DFS (HR 1.09 [95% CI 0.50, 2.40], p=0.829) (Table 2, Fig 3). Detecting true statistical significance was limited due to the small number of patients within each treatment arm. However, despite this, patients treated with the combination of urelumab, nivolumab, and Cy-GVAX (Arm C) demonstrated a clinically-compelling benefit in DFS when compared against those treated with Cy-GVAX alone (HR 0.55[95% CI 0.21,1.49],p=0.242) or Cy-GVAX with nivolumab (HR 0.51[95% CI 0.19,1.35],p=0.173) (Table 2, Fig 3). This trend persisted after controlling for age, nodal spread, and adjuvant chemotherapy regimen (HR=0.64 [95% CI 0.19-2.19], p=0.478 compared with Arm A; HR=0.48 [95% CI 0.15-1.60], p=0.232 compared with Arm B) (Table S1).	respectively (Table 2, Fig 3). Detecting true statistical significance was limited due to the small number of patients within each treatment arm. In context of this, compared to Cy-GVAX alone (Arm A), adding nivolumab to Cy-GVAX (Arm B) did not improve DFS (HR 1.09 [95% CI 0.50, 2.40], p=0.829) (Table 2, Fig 3). Patients treated with the combination of urelumab, nivolumab, and Cy-GVAX (Arm C) demonstrated numerically-improved DFS when compared against those treated with Cy-GVAX alone (HR 0.55 [95% CI 0.21,1.49],p=0.242) or Cy-GVAX with nivolumab (HR 0.51 [95% CI 0.19,1.35],p=0.173) (Table 2, Fig 3), but did not reach statistical significance. This favorable HR trend, though again not statistically significant, persisted after controlling for age, nodal spread, and adjuvant chemotherapy regimen (HR=0.64 [95% CI 0.19-2.19], p=0.478 compared with Arm A; HR=0.48 [95% CI 0.15-1.60], p=0.232 compared with Arm B) (Table S1).
---	--

Original	Revised
Median OS (95% CI) was 23.59 mo (13.27, NR), 27.01 mo (20.76, NR), and 35.55 mo (17.74, NR) for Arms A, B, C, respectively (Table 2, Fig 3). Compared to Cy-GVAX alone, adding PD1 to Cy-GVAX did not improve OS (HR=1.11 [95% CI 0.47, 2.63], p=0.813) (Table 2, Fig 3). Patients treated with the combination of CD137 + PD1 + Cy-GVAX showed a clinically meaningful improvement in OS when compared against those treated with Cy-GVAX alone (HR 0.59 [95% CI 0.18, 1.91], p=0.377) and in combination PD1 (HR=0.53 [95% CI 0.17, 1.67], p=0.279) (Table 2, Fig 3). Similar to DFS, this clinically-meaningful HR persisted after controlling for age, nodal spread, and adjuvant chemotherapy regimen (HR=0.41 [95% CI 0.10-1.62], p=0.202 compared to Arm A; HR=0.59 (95% CI 0.18-1.91), p=0.377 compared to Arm B) (Table 2). On MVA, presence of nodal spread at time of	Median OS (95% CI) was 23.59 mo (13.27, NR), 27.01 mo (20.76, NR), and 35.55 mo (17.74, NR) for Arms A, B, C, respectively (Table 2, Fig 3). Compared to Cy-GVAX alone, adding PD1 to Cy-GVAX did not improve OS (HR=1.11 [95% CI 0.47, 2.63], p=0.813) (Table 2, Fig 3). Patients treated with the combination of CD137 + PD1 + Cy-GVAX showed a numerically-improved OS when compared against those treated with Cy-GVAX alone (HR 0.59 [95% CI 0.18, 1.91], p=0.377) and in combination PD1 (HR=0.53 [95% CI 0.17, 1.67], p=0.279) (Table 2, Fig 3), but did not reach statistical significance. Similar to DFS, this favorable HR persisted after controlling for age, nodal spread, and adjuvant chemotherapy regimen (HR=0.75 [95% CI 0.18-3.10], p=0.0.692 compared to Arm A; HR=0.41 (95% CI 0.10-1.62), p=0.202 compared to Arm B) (Table 2), but did not reach statistical significance.

surgery correlated with worse OS (HR=2.92 [1.02-8.32], p=0.045) and trended towards worse DFS (HR=2.21 [0.88-5.53], p=0.091) Table S1, Table S2). Type of SOC adjuvant systemic treatment was not significantly correlated with DFS or OS in our study sample (Table S1, Table S2, Fig S3).	On MVA, presence of nodal spread at time of surgery correlated with worse OS (HR=2.92 [1.02-8.32], p=0.045) and trended towards worse DFS (HR=2.21 [0.88-5.53], p=0.091) Table S1, Table S2). Type of SOC adjuvant systemic treatment was not significantly correlated with DFS or OS in our study sample nor was tumor-stage (Table S1, Table S2, Fig S3).
--	--

11. In the abstract, “combination Cy-GVAX+nivolumab+urelumab demonstrated a clinically-meaningful improved DFS compared to Cy-GVAX alone” “(HR 0.55[95%CI 0.21,149],p=0.242)”

I formally don't disagree looks intriguing. But these statements are not statistically supported.

Author reply: Thank you for the feedback. We have made the following revisions to address this comment:

Original	Revised
Results: Forty patients (n=16[A],n=14[B],n=10[C]) were eligible for efficacy analysis. Median DFS(95%CI) was 13.90mo(5.59,NR), 14.98mo(7.95,44.09) and 33.51(16.76,NR) for Arms A/B/C, respectively. Combination Cy-GVAX+nivolumab+urelumab demonstrated a clinically-meaningful improved DFS compared to Cy-GVAX alone (HR 0.55[95%CI 0.21,149],p=0.242) and Cy-GVAX+nivolumab (HR 0.51[95%CI 0.19,1.35],p=0.173). All three treatments were well tolerated. Treatment with combination Cy-GVAX+nivolumab+urelumab met the primary endpoint by significantly increasing intratumoral CD8+CD137+ and CD8+CD137+GZMB+ T cells compared to Cy-GVAX± Nivolumab treatment.	Results: Forty patients (n=16[A],n=14[B],n=10[C]) were eligible for efficacy analysis. Treatment with combination Cy-GVAX+nivolumab+urelumab met the primary endpoint by significantly increasing intratumoral CD8+CD137+ and CD8+CD137+GZMB+ T cells compared to Cy-GVAX± Nivolumab treatment. Median DFS(95%CI) was 13.90mo(5.59,NR), 14.98mo(7.95,44.09) and 33.51(16.76,NR) for Arms A,B,C, respectively. While the combination Cy-GVAX+nivolumab+urelumab demonstrated numerically-improved DFS compared to Cy-GVAX alone (HR 0.55[95%CI 0.21,1.49],p=0.242) and Cy-GVAX+nivolumab (HR 0.51[95%CI 0.19,1.35],p=0.173), this was not statistically significant and was limited by a small sample size as well as imbalance in standard of care adjuvant chemotherapy regimens. All three treatments were well tolerated.

Minor Comments.

1. The authors should consider, where possible, simplify or clarify the table in figure 2. It is redundant with some of the text and a bit burdensome. (This is minor)

Author reply: We appreciate the reviewer’s feedback. We have chosen to leave the CONSORT diagram as is because the checkpoints listed are important for understanding the design/schema and also to detail where attrition took place (and the reasons for this). Since this was listed as minor comment, we hope the reviewer will be open to our reasoning.

2. Figure 3 could use some improvement in layout. Y axis are very far from DFS/OS curves.

Author reply: Thank you for the suggestion. We have modified figures as suggested.

3. The images in Fig 4d-e are not the best.

Author reply: Thank you for the comment. We also realize that the figures had lost the resolutions when they were converted to the PDF format. We have re-made the figure (**now Figure 3**) in a high resolution.

Reviewer #3 (Remarks to the Author): with expertise in pancreatic cancer, immunotherapy

1. Neoadjuvant trials with short-course, upfront immunotherapy are advantageous for looking at

tissue endpoints in resected specimens, and it removes the confounding effects of cytotoxic chemotherapy, so in that way the design is seen as a strength.

Author reply: Thank you for your comment and mentioning the strength of the concept and purpose of the platform trial

2. While the concept of platform trials is interesting, it is not novel and has been used and published in other cancer types.

Author reply: We agree that the concept of platform trials is not novel as it has been used in other cancer types. Indeed, as one of the first for solid tumors, our neoadjuvant platform was first initiated in 2008 (Zheng et al. Clinical Cancer Research 2020; Lutz et al. Cancer Immunology Research 2014). The biological endpoint design in our platform trial may also be considered to be novel.

3. GVAX has been around for a long time and has not shown to improve clinically meaningful outcomes, even with the addition of nivolumab.

Author reply: We agree with Reviewer. We did not see a clinically meaningful improvement of outcome with GVAX alone or GVAX in combination with nivolumab. Therefore, we were very excited by the results with the addition of urelumab in Arm C. We recognize that the small sample size in Arm C has limited the conclusions that could be drawn from this study; however, this study has provided the clinical efficacy and immune response signals rapidly with only a small number of patients tested. Therefore, using a small sample size is also a “strength” of this study. We are currently planning a randomized phase 2 study to confirm the efficacy of the triple combination in Arm C.

4. While the results of the tissue analysis are interesting, the survival results of the triplet therapy are not statistically significant, and the patient numbers are too small to make any meaningful efficacy conclusions.

Author reply: Thank you for the feedback. We agree that the small sample size in Arm C has limited the conclusions that could be drawn from this study; however, this study has provided the clinical efficacy and immune response signals rapidly with only a small number of patients tested. We are currently planning a randomized phase 2 study designed (and powered) to evaluate the clinical efficacy of the triple combination in Arm C. We have made the following revisions to address this comment:

Original	Revised
To address a potential confounder, the higher percentage of Arm C patients receiving adjuvant (m)FOLFIRINOX compared to patients in Arms A & B, multiple strategies were employed to evaluate the additive contribution of IO triplet combination to the observed DFS trends. The survival hazard models attempted to control for chemo	It is important to acknowledge the higher percentage of Arm C patients receiving adjuvant (m)FOLFIRINOX compared to patients in Arms A and B. To address this potential confounder, multiple strategies were employed to evaluate the additive contribution of IO triplet combination to the observed DFS trends. The multivariable

regimens (Table S1, Table S2). Additionally, across all treatment arms, patients who received adjuvant (m)FOLFIRINOX appeared to have similar DFS when compared to the collective study participants who were treated with gemcitabine-based regimens (Fig S3). Finally, Arm C patients were compared against a historical control cohort of resected PDA treated at Johns Hopkins Sidney Kimmel Cancer Center during the time of Arm C's enrollment. When matched 3:1 on adjuvant chemo regimen, age, and nodal disease status with propensity score matching (Table S5, Fig S7), Arm C patients maintained a favorable DFS HR: Arm C mDFS = 33.02 mo; Historical Control mDFS= 20.83 mo; stratified HR 0.72 [0.29-1.80], p=0.480 (*DFS was measured starting the day of surgery for both groups) (Table S6, Fig S7). It should be noted that the historical cohort's DFS carries a potential lead-time bias due to the follow up and restaging scan schedule being more stringent for patients on the trial. Even with the increased matching ratio, the sample size remains modest for comparison. However, the notable difference in median DFS, and visible separation on the survival curves, argue for a larger, follow up phase II trial powered for clinical outcomes with uniform IO dosing, and SOC adjuvant regimens.

survival analysis attempted to control for chemo regimen. Additionally, across all treatment arms, patients who received adjuvant (m)FOLFIRINOX appeared to have similar DFS when compared to the collective study participants who were treated with gemcitabine-based regimens. Finally, Arm C patients were also compared against a matched-historical control cohort. There are clear limitations of this study and the above analyses driven largely by the sample size. However, the early clinical and immune response signals observed in this small cohort support a follow up, randomized, phase 2 study designed and powered to assess the clinical efficacy of the triple IO combination used in Arm C.

5. Post-op adjuvant chemo +/- XRT was not standardized, a potential confounder of the efficacy results.

Author reply: We agree with that this is limitation. Multiple efforts were made to address this potential confounder as described in the results and discussion sections including multivariate analysis of the study cohort as well as generating a matched-historical control cohort for comparison. We acknowledge that even with these measures, the small overall numbers do not allow for adequate control of this variable. A future study that is powered to assess clinical efficacy should be designed with uniform adjuvant SOC treatment regimens. We revised the manuscript as described above in the Reply to Comment #4.

6. What was the rationale for the “extended treatment Phase with nivolumab and does this amended change during the protocol further confound the results?

Author reply: The extended treatment phase, added as part of an amendment approved by the IRB in 2018, was in response to a change in standard dosing strategy for immune checkpoint inhibition use in the adjuvant setting based on published data supporting 1 year adjuvant immunotherapy for in other solid tumors such as melanoma (CheckMate 238 [published in 2017], EORTC 1325 [published in 2018]). In terms of whether this was a potential confounder, since Urelumab was not available to be continued in the extended phase treatment setting, and the lack of added benefit observed with adding NIVO alone to cy-GVAX, it seems less likely the extended treatment phase significantly confounded outcomes. It is our hope that these results will lend support to a future phase II trial, powered for clinical outcomes, that will be designed with uniform immunomodulator dosing and duration.

7. 30% of the patients in ARM C had T1 tumors (compared with 14-18% in A and B) which alone could account for the survival advantage seen in patients in the triplet therapy ARM.

Author reply: We agree that there were numerical differences in T1 stage disease but T-stage did not correlate with PFS/OS in this particular study. However, presence/absence of nodal disease did correlate with survival and the rates of nodal positive disease were similar between Arms C [70%] and A [68.8%] & B [71.4%]. We also matched on nodal disease when comparing to the historical control cohort.

Reviewer #4 (Remarks to the Author): with expertise in biostatistics, clinical trial study design

In this paper, the authors conducted a three-arms platform trial to demonstrate the feasibility of testing novel immunotherapy combinations in patients with resectable PDA. The primary endpoints were survival outcomes and immune endpoints. I have the following questions regarding to the statistical design and analysis:

1. Due to the small sample size, it is hard to tell that the HR for survival outcome demonstrate real clinical benefit. None of the p-values for HR is significant (<0.05), and the 95% upper quantiles for most HR are far beyond 1. More patients are needed to confirm the finding.

Author reply: We agree. Clinical outcomes were not the primary outcomes. The secondary clinical outcomes have such been deemphasized in the revised manuscript.

2. The description of sample size consideration is very unclear. The pre-specified effect size is huge, do you have any data to support such setting? What test do you used for power calculation? Do you consider the multiple comparison for sample size?

Author reply: We apologize for the lack of clarity. We recognize it would be difficult for readers to find this information in the attached clinical protocol, particularly in such a complex

platform clinical trial protocol. We therefore have added it in the revised manuscript. The effect size was projected based on preliminary data of early patients in Arm A and B (Li et al., 2022, Cancer Cell). We did not consider multiple comparisons because each of the comparisons (IL-17A expression and CD8+CD137+ cell density) were of interest. We clarified in the revised manuscript: *Since both primary biologic endpoints - 1) comparing IL17A expression between Arm A and B, and 2) comparing CD137+ T cell density between Arm C and B - were each of respective interest, they were not subjected to the multiple comparison adjustment.*

3. The endpoints are inconsistent across different arms. “The primary endpoint for Arms A and B was IL17A expression in vaccine-induced lymphoid aggregates in resected PDAs from patients treated with the combination of Cy-GVAX with or without nivolumab (19). The primary biologic endpoint for Arm C was CD8+CD137+T cell density within tumor regions of interest (containing at least one TLA) in surgically resected specimens.” This should cause problems for statistical testing between Arm A/B and Arm C.

Author reply: Thank you for the feedback. For the platform trial design, arms have been added sequentially. This platform trial initially has two arms, Arm A and Arm B; therefore, Arm A and Arm B had a different primary endpoint. With the three arms in this report, Arm C was added later based on the results of the correlative studies of Arm A and B. Therefore, the biological endpoint for Arm C was to assess the targeted effect on CD8+CD137+T cell with the addition of urelumab, a CD137 agonist mAb, to the combination of cy-GVAX + nivolumab. The manuscript includes discussions of the limitations of comparisons between noncontemporary arms.

4. Why the non-parametric analysis is used for immune outcome, not the t-test? Do you check the normality of the data? Better provide the Q-Q plot.

Author reply: Thank you for the feedback. We felt that the assessment of normality based on observed data may not be reliable especially with the small sample size. Therefore, we applied non-parametric analysis, as it did not rely on the assumption of normality of data for the results to be valid.

5. Line 181 [Abstract], “Cy-GVAX alone (HR 0.55[95%CI 0.21,149],p=0.242)”, should 149 be 1.49? Also, why the results (p-values) for the immune endpoints are not mentioned here?

Author reply: Thank you for pointing out the error. Yes, 149 should be 1.49. This has been corrected in the abstract as well as adding in the relevant p-values requested.

REVIEWERS' COMMENTS	AUTHOR RESPONSE
Reviewer #1 (Remarks to the Author):	
Thank you to the authors for the detailed response to reviewer comments. As noted in the initial review, this is an interesting and innovative trial, and overall well written manuscript. The main issues raised in the initial review related to the over emphasis of the efficacy data, given the limitations of small sample size and in-balance between arms. The authors have addressed all of my comments/suggestions, and the limitations of the efficacy results are more clearly stated.	Thank you for these comments. We very much appreciate the feedback and the opportunity to strengthen our work. We are glad to hear that our revisions and responses are to your satisfaction.
Reviewer #2 (Remarks to the Author):	
I thank the authors for their responses and the paper is improved. I do really like this study and recognize the importance of the question. That said the response(s) leave me challenged to be a champion for this paper. My main challenge I had to prior version was to push the biomarker part a bit further and to provide some insight into the immunologic effects of TIGIT in the combination. If I correctly understood the responses; a further in depth analysis of Arm C (with Tight) is being held for another paper. This paper was provided and also looks nice, but does not change the current manuscript and what is in it. The authors did improve the analysis of the currently finished multiplex IHC as requested (thank you, it looks great). But I find the choice to not push this papers biomarker studies any further (even with a second focused IHC study), makes it really difficult for me to champion this paper alone. And I would otherwise love to. So, the authors have made an attempt to address my concerns, without adding new experiments (because of the above). I have no outstanding concerns for the data presented other than it limits in scope. I will thus leave the balance of the decision to the other	We very much appreciate the feedback and the opportunity to strengthen our work. Thank you for these comments. We understand that Reviewer #2 hopes we push biomarker studies further, particularly, on TIGIT, but is overall satisfied by our revised manuscript. We appreciate his/her suggestion. We indeed plan to have a more in-depth analysis on the T cell exhaustion pathway down the road although we haven't focused on TIGIT for this manuscript, TIGIT IHC staining was newly added in response to Reviewer's prior comments. It would take some time to complete a second focused study on the TIGIT pathway. In addition to the TIGIT IHC staining (Supplementary Figure 9, Results Subsection: Exploratory Immune Analysis, paragraph #2) we have added the following sentence to our discussion that discusses next steps in biomarker analysis (including examining T cell exhaustion pathways) as one of the future directions for this platform trial and its correlative studies: "Additional biomarker studies are warranted, particularly on the immunosuppressive TME and T cell exhaustion pathways, to inform new Arm design for our platform trial."

reviewers and editor. Maybe the clinical merits can carry the day. One minor point. 1. Figure 3A all colors bleed to white, so can't see red or yellow or green T cells. Suggest choosing different false colors or splitting the colors into two images, either may make the image more obvious.	We appreciate this comment and have chosen different pseudocolors in Fig3a for better discernment.
Reviewer #4 (Remarks to the Author):	
The authors have addressed all of my comments and I have no more questions.	We are glad to hear that our revisions and responses are to your satisfaction.